# N-BEATS: NEURAL BASIS EXPANSION ANALYSIS FOR INTERPRETABLE TIME SERIES FORECASTING

**Boris N. Oreshkin**
Element AI
boris.oreshkin@gmail.com

**Dmitri Carpov**
Element AI
dmitri.carpov@elementai.com

**Nicolas Chapados**
Element AI
chapados@elementai.com

**Yoshua Bengio**
Mila
yoshua.bengio@mila.quebec

## ABSTRACT

We focus on solving the univariate times series point forecasting problem using deep learning. We propose a deep neural architecture based on backward and forward residual links and a very deep stack of fully-connected layers. The architecture has a number of desirable properties, being interpretable, applicable without modification to a wide array of target domains, and fast to train. We test the proposed architecture on several well-known datasets, including M3, M4 and TOURISM competition datasets containing time series from diverse domains. We demonstrate state-of-the-art performance for two configurations of N-BEATS for all the datasets, improving forecast accuracy by 11% over a statistical benchmark and by 3% over last year's winner of the M4 competition, a domain-adjusted hand-crafted hybrid between neural network and statistical time series models. The first configuration of our model does not employ any time-series-specific components and its performance on heterogeneous datasets strongly suggests that, contrarily to received wisdom, deep learning primitives such as residual blocks are by themselves sufficient to solve a wide range of forecasting problems. Finally, we demonstrate how the proposed architecture can be augmented to provide outputs that are interpretable without considerable loss in accuracy.

## 1 INTRODUCTION

Time series (TS) forecasting is an important business problem and a fruitful application area for machine learning (ML). It underlies most aspects of modern business, including such critical areas as inventory control and customer management, as well as business planning going from production and distribution to finance and marketing. As such, it has a considerable financial impact, often ranging in the millions of dollars for every point of forecasting accuracy gained (Jain, 2017; Kahn, 2003). And yet, unlike areas such as computer vision or natural language processing where deep learning (DL) techniques are now well entrenched, there still exists evidence that ML and DL struggle to outperform classical statistical TS forecasting approaches (Makridakis et al., 2018a;b). For instance, the rankings of the six "pure" ML methods submitted to M4 competition were 23, 37, 38, 48, 54, and 57 out of a total of 60 entries, and most of the best-ranking methods were ensembles of classical statistical techniques (Makridakis et al., 2018b).

On the other hand, the M4 competition winner (Smyl, 2020), was based on a hybrid between neural residual/attention dilated LSTM stack with a classical Holt-Winters statistical model (Holt, 1957; 2004; Winters, 1960) with learnable parameters. Since Smyl's approach heavily depends on this Holt-Winters component, Makridakis et al. (2018b) further argue that "hybrid approaches and combinations of method are the way forward for improving the forecasting accuracy and making forecasting more valuable". In this work we aspire to challenge this conclusion by exploring the potential of pure DL architectures in the context of the TS forecasting. Moreover, in the context of interpretable DL architecture design, we are interested in answering the following question: can we

inject a suitable inductive bias in the model to make its internal operations more interpretable, in the sense of extracting some explainable driving factors combining to produce a given forecast?

## 1.1 SUMMARY OF CONTRIBUTIONS

**Deep Neural Architecture:** To the best of our knowledge, this is the first work to empirically demonstrate that pure DL using no time-series specific components outperforms well-established statistical approaches on M3, M4 and TOURISM datasets (on M4, by 11% over statistical benchmark, by 7% over the best statistical entry, and by 3% over the M4 competition winner). In our view, this provides a long-missing proof of concept for the use of pure ML in TS forecasting and strengthens motivation to continue advancing the research in this area.

**Interpretable DL for Time Series:** In addition to accuracy benefits, we also show that it is feasible to design an architecture with interpretable outputs that can be used by practitioners in very much the same way as traditional decomposition techniques such as the "seasonality-trend-level" approach (Cleveland et al., 1990).

## 2 PROBLEM STATEMENT

We consider the univariate point forecasting problem in discrete time. Given a length-$H$ forecast horizon a length-$T$ observed series history $[y_1, \ldots, y_T] \in \mathbb{R}^T$, the task is to predict the vector of future values $\mathbf{y} \in \mathbb{R}^H = [y_{T+1}, y_{T+2}, \ldots, y_{T+H}]$. For simplicity, we will later consider a *lookback window* of length $t \leq T$ ending with the last observed value $y_T$ to serve as model input, and denoted $\mathbf{x} \in \mathbb{R}^t = [y_{T-t+1}, \ldots, y_T]$. We denote $\widehat{\mathbf{y}}$ the forecast of $\mathbf{y}$. The following metrics are commonly used to evaluate forecasting performance (Hyndman & Koehler, 2006; Makridakis & Hibon, 2000; Makridakis et al., 2018b; Athanasopoulos et al., 2011):

$$\text{sMAPE} = \frac{200}{H} \sum_{i=1}^{H} \frac{|y_{T+i} - \widehat{y}_{T+i}|}{|y_{T+i}| + |\widehat{y}_{T+i}|}, \qquad \text{MAPE} = \frac{100}{H} \sum_{i=1}^{H} \frac{|y_{T+i} - \widehat{y}_{T+i}|}{|y_{T+i}|},$$

$$\text{MASE} = \frac{1}{H} \sum_{i=1}^{H} \frac{|y_{T+i} - \widehat{y}_{T+i}|}{\frac{1}{T+H-m} \sum_{j=m+1}^{T+H} |y_j - y_{j-m}|}, \qquad \text{OWA} = \frac{1}{2} \left[ \frac{\text{sMAPE}}{\text{sMAPE}_{\text{Naïve2}}} + \frac{\text{MASE}}{\text{MASE}_{\text{Naïve2}}} \right].$$

Here $m$ is the periodicity of the data (*e.g.*, 12 for monthly series). MAPE (Mean Absolute Percentage Error), sMAPE (symmetric MAPE) and MASE (Mean Absolute Scaled Error) are standard scale-free metrics in the practice of forecasting (Hyndman & Koehler, 2006; Makridakis & Hibon, 2000): whereas sMAPE scales the error by the average between the forecast and ground truth, the MASE scales by the average error of the naïve predictor that simply copies the observation measured $m$ periods in the past, thereby accounting for seasonality. OWA (overall weighted average) is a M4-specific metric used to rank competition entries (M4 Team, 2018b), where sMAPE and MASE metrics are normalized such that a seasonally-adjusted naïve forecast obtains OWA = 1.0.

## 3 N-BEATS

Our architecture design methodology relies on a few key principles. First, the base architecture should be simple and generic, yet expressive (deep). Second, the architecture should not rely on time-series-specific feature engineering or input scaling. These prerequisites let us explore the potential of pure DL architecture in TS forecasting. Finally, as a prerequisite to explore interpretability, the architecture should be extendable towards making its outputs human interpretable. We now discuss how those principles converge to the proposed architecture.

### 3.1 BASIC BLOCK

The proposed basic building block has a fork architecture and is depicted in Fig. 1 (left). We focus on describing the operation of $\ell$-th block in this section in detail (note that the block index $\ell$ is dropped in Fig. 1 for brevity). The $\ell$-th block accepts its respective input $\mathbf{x}_\ell$ and outputs two vectors, $\widehat{\mathbf{x}}_\ell$ and $\widehat{\mathbf{y}}_\ell$. For the very first block in the model, its respective $\mathbf{x}_\ell$ is the overall model input — a history lookback window of certain length ending with the last measured observation. We set the length of

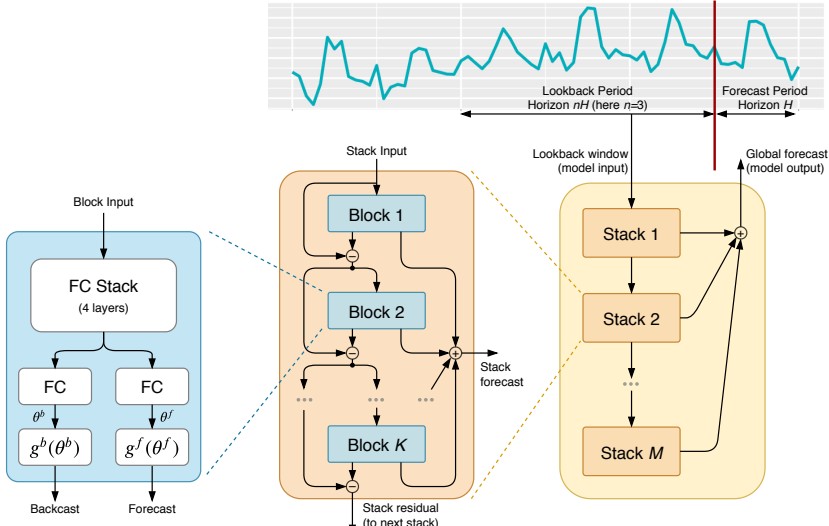

Figure 1: Proposed architecture. The basic building block is a multi-layer FC network with RELU nonlinearities. It predicts basis expansion coefficients both forward, $\theta^f$, (forecast) and backward, $\theta^b$, (backcast). Blocks are organized into stacks using doubly residual stacking principle. A stack may have layers with shared $g^b$ and $g^f$. Forecasts are aggregated in hierarchical fashion. This enables building a very deep neural network with interpretable outputs.

input window to a multiple of the forecast horizon $H$, and typical lengths of $\mathbf{x}$ in our setup range from $2H$ to $7H$. For the rest of the blocks, their inputs $\mathbf{x}_\ell$ are residual outputs of the previous blocks. Each block has two outputs: $\widehat{\mathbf{y}}_\ell$, the block's forward forecast of length $H$; and $\widehat{\mathbf{x}}_\ell$, the block's best estimate of $\mathbf{x}_\ell$, also known as the 'backcast', given the constraints on the functional space that the block can use to approximate signals.

Internally, the basic building block consists of two parts. The first part is a fully connected network that produces the forward $\theta_\ell^f$ and the backward $\theta_\ell^b$ predictors of expansion coefficients (again, note that the block index $\ell$ is dropped for $\theta_\ell^b$, $\theta_\ell^f$, $g_\ell^b$, $g_\ell^f$ in Fig. 1 for brevity). The second part consists of the backward $g_\ell^b$ and the forward $g_\ell^f$ basis layers that accept the respective forward $\theta_\ell^f$ and backward $\theta_\ell^b$ expansion coefficients, project them internally on the set of basis functions and produce the backcast $\widehat{\mathbf{x}}_\ell$ and the forecast outputs $\widehat{\mathbf{y}}_\ell$ defined in the previous paragraph.

The operation of the first part of the $\ell$-th block is described by the following equations:

$$\mathbf{h}_{\ell,1} = \text{FC}_{\ell,1}(\mathbf{x}_\ell), \quad \mathbf{h}_{\ell,2} = \text{FC}_{\ell,2}(\mathbf{h}_{\ell,1}), \quad \mathbf{h}_{\ell,3} = \text{FC}_{\ell,3}(\mathbf{h}_{\ell,2}), \quad \mathbf{h}_{\ell,4} = \text{FC}_{\ell,4}(\mathbf{h}_{\ell,3}).$$
$$\theta_\ell^b = \text{LINEAR}_\ell^b(\mathbf{h}_{\ell,4}), \quad \theta_\ell^f = \text{LINEAR}_\ell^f(\mathbf{h}_{\ell,4}). \tag{1}$$

Here LINEAR layer is simply a linear projection layer, i.e. $\theta_\ell^f = \mathbf{W}_\ell^f \mathbf{h}_{\ell,4}$. The FC layer is a standard fully connected layer with RELU non-linearity (Nair & Hinton, 2010), such that for $\text{FC}_{\ell,1}$ we have, for example: $\mathbf{h}_{\ell,1} = \text{RELU}(\mathbf{W}_{\ell,1}\mathbf{x}_\ell + \mathbf{b}_{\ell,1})$. One task of this part of the architecture is to predict the forward expansion coefficients $\theta_\ell^f$ with the ultimate goal of optimizing the accuracy of the partial forecast $\widehat{\mathbf{y}}_\ell$ by properly mixing the basis vectors supplied by $g_\ell^f$. Additionally, this sub-network predicts backward expansion coefficients $\theta_\ell^b$ used by $g_\ell^b$ to produce an estimate of $\mathbf{x}_\ell$ with the ultimate goal of helping the downstream blocks by removing components of their input that are not helpful for forecasting.

The second part of the network maps expansion coefficients $\theta_\ell^f$ and $\theta_\ell^b$ to outputs via basis layers, $\widehat{\mathbf{y}}_\ell = g_\ell^f(\theta_\ell^f)$ and $\widehat{\mathbf{x}}_\ell = g_\ell^b(\theta_\ell^b)$. Its operation is described by the following equations:

$$\widehat{\mathbf{y}}_\ell = \sum_{i=1}^{\dim(\theta_\ell^f)} \theta_{\ell,i}^f \mathbf{v}_i^f, \quad \widehat{\mathbf{x}}_\ell = \sum_{i=1}^{\dim(\theta_\ell^b)} \theta_{\ell,i}^b \mathbf{v}_i^b.$$

Here $\mathbf{v}_i^f$ and $\mathbf{v}_i^b$ are forecast and backcast basis vectors, $\theta_{\ell,i}^f$ is the $i$-th element of $\theta_\ell^f$. The function of $g_\ell^b$ and $g_\ell^f$ is to provide sufficiently rich sets $\{\mathbf{v}_i^f\}_{i=1}^{\dim(\theta_\ell^f)}$ and $\{\mathbf{v}_i^b\}_{i=1}^{\dim(\theta_\ell^b)}$ such that their respective outputs can be represented adequately via varying expansion coefficients $\theta_\ell^f$ and $\theta_\ell^b$. As shown below, $g_\ell^b$ and $g_\ell^f$ can either be chosen to be learnable or can be set to specific functional forms to reflect certain problem-specific inductive biases in order to appropriately constrain the structure of outputs. Concrete examples of $g_\ell^b$ and $g_\ell^f$ are discussed in Section 3.3.

## 3.2 Doubly Residual Stacking

The classical residual network architecture adds the input of the stack of layers to its output before passing the result to the next stack (He et al., 2016). The DenseNet architecture proposed by Huang et al. (2017) extends this principle by introducing extra connections from the output of each stack to the input of every other stack that follows it. These approaches provide clear advantages in improving the trainability of deep architectures. Their disadvantage in the context of this work is that they result in network structures that are difficult to interpret. We propose a novel hierarchical doubly residual topology depicted in Fig. 1 (middle and right). The proposed architecture has two residual branches, one running over backcast prediction of each layer and the other one is running over the forecast branch of each layer. Its operation is described by the following equations:

$$\mathbf{x}_\ell = \mathbf{x}_{\ell-1} - \widehat{\mathbf{x}}_{\ell-1}, \quad \widehat{\mathbf{y}} = \sum_\ell \widehat{\mathbf{y}}_\ell.$$

As previously mentioned, in the special case of the very first block, its input is the model level input $\mathbf{x}$, $\mathbf{x}_1 \equiv \mathbf{x}$. For all other blocks, the backcast residual branch $\mathbf{x}_\ell$ can be thought of as running a sequential analysis of the input signal. Previous block removes the portion of the signal $\widehat{\mathbf{x}}_{\ell-1}$ that it can approximate well, making the forecast job of the downstream blocks easier. This structure also facilitates more fluid gradient backpropagation. More importantly, each block outputs a partial forecast $\widehat{\mathbf{y}}_\ell$ that is first aggregated at the stack level and then at the overall network level, providing a hierarchical decomposition. The final forecast $\widehat{\mathbf{y}}$ is the sum of all partial forecasts. In a generic model context, when stacks are allowed to have arbitrary $g_\ell^b$ and $g_\ell^f$ for each layer, this makes the network more transparent to gradient flows. In a special situation of deliberate structure enforced in $g_\ell^b$ and $g_\ell^f$ shared over a stack, explained next, this has the critical importance of enabling interpretability via the aggregation of meaningful partial forecasts.

## 3.3 Interpretability

We propose two configurations of the architecture, based on the selection of $g_\ell^b$ and $g_\ell^f$. One of them is generic DL, the other one is augmented with certain inductive biases to be interpretable.

The **generic architecture** does not rely on TS-specific knowledge. We set $g_\ell^b$ and $g_\ell^f$ to be a linear projection of the previous layer output. In this case the outputs of block $\ell$ are described as:

$$\widehat{\mathbf{y}}_\ell = \mathbf{V}_\ell^f \theta_\ell^f + \mathbf{b}_\ell^f, \quad \widehat{\mathbf{x}}_\ell = \mathbf{V}_\ell^b \theta_\ell^b + \mathbf{b}_\ell^b.$$

The interpretation of this model is that the FC layers in the basic building block depicted in Fig. 1 learn the predictive decomposition of the partial forecast $\widehat{\mathbf{y}}_\ell$ in the basis $\mathbf{V}_\ell^f$ learned by the network. Matrix $\mathbf{V}_\ell^f$ has dimensionality $H \times \dim(\theta_\ell^f)$. Therefore, the first dimension of $\mathbf{V}_\ell^f$ has the interpretation of discrete time index in the forecast domain. The second dimension of the matrix has the interpretation of the indices of the basis functions, with $\theta_\ell^f$ being the expansion coefficients for this basis. Thus the columns of $\mathbf{V}_\ell^f$ can be thought of as waveforms in the time domain. Because no additional constraints are imposed on the form of $\mathbf{V}_\ell^f$, the waveforms learned by the deep model do not have inherent structure (and none is apparent in our experiments). This leads to $\widehat{\mathbf{y}}_\ell$ not being interpretable.

The **interpretable architecture** can be constructed by reusing the overall architectural approach in Fig. 1 and by adding structure to basis layers at stack level. Forecasting practitioners often use the decomposition of time series into trend and seasonality, such as those performed by the STL (Cleveland et al., 1990) and X13-ARIMA (U.S. Census Bureau, 2013). We propose to design the trend and seasonality decomposition into the model to make the stack outputs more easily interpretable. Note

that for the generic model the notion of stack was not necessary and the stack level indexing was omitted for clarity. Now we will consider both stack level and block level indexing. For example, $\widehat{\mathbf{y}}_{s,\ell}$ will denote the partial forecast of block $\ell$ within stack $s$.

**Trend model.** A typical characteristic of trend is that most of the time it is a monotonic function, or at least a slowly varying function. In order to mimic this behaviour we propose to constrain $g_{s,\ell}^b$ and $g_{s,\ell}^f$ to be a polynomial of small degree $p$, a function slowly varying across forecast window:

$$\widehat{\mathbf{y}}_{s,\ell} = \sum_{i=0}^{p} \theta_{s,\ell,i}^f t^i. \tag{2}$$

Here time vector $\mathbf{t} = [0, 1, 2, \dots, H-2, H-1]^T / H$ is defined on a discrete grid running from 0 to $(H-1)/H$, forecasting $H$ steps ahead. Alternatively, the trend forecast in matrix form will then be:

$$\widehat{\mathbf{y}}_{s,\ell}^{tr} = \mathbf{T} \theta_{s,\ell}^f,$$

where $\theta_{s,\ell}^f$ are polynomial coefficients predicted by a FC network of layer $\ell$ of stack $s$ described by equations (1); and $\mathbf{T} = [\mathbf{1}, \mathbf{t}, \dots, \mathbf{t}^p]$ is the matrix of powers of $\mathbf{t}$. If $p$ is low, e.g. 2 or 3, it forces $\widehat{\mathbf{y}}_{s,\ell}^{tr}$ to mimic trend.

**Seasonality model.** Typical characteristic of seasonality is that it is a regular, cyclical, recurring fluctuation. Therefore, to model seasonality, we propose to constrain $g_{s,\ell}^b$ and $g_{s,\ell}^f$ to belong to the class of periodic functions, *i.e.* $y_t = y_{t-\Delta}$, where $\Delta$ is a seasonality period. A natural choice for the basis to model periodic function is the Fourier series:

$$\widehat{\mathbf{y}}_{s,\ell} = \sum_{i=0}^{\lfloor H/2-1 \rfloor} \theta_{s,\ell,i}^f \cos(2\pi i t) + \theta_{s,\ell,i+\lfloor H/2 \rfloor}^f \sin(2\pi i t), \tag{3}$$

The seasonality forecast will then have the matrix form as follows:

$$\widehat{\mathbf{y}}_{s,\ell}^{seas} = \mathbf{S} \theta_{s,\ell}^f,$$

where $\theta_{s,\ell}^f$ are Fourier coefficients predicted by a FC network of layer $\ell$ of stack $s$ described by equations (1); and $\mathbf{S} = [\mathbf{1}, \cos(2\pi \mathbf{t}), \dots \cos(2\pi \lfloor H/2-1 \rfloor \mathbf{t})), \sin(2\pi \mathbf{t}), \dots, \sin(2\pi \lfloor H/2-1 \rfloor \mathbf{t}))]$ is the matrix of sinusoidal waveforms. The forecast $\widehat{\mathbf{y}}_{s,\ell}^{seas}$ is then a periodic function mimicking typical seasonal patterns.

The overall interpretable architecture consists of two stacks: the trend stack is followed by the seasonality stack. The doubly residual stacking combined with the forecast/backcast principle result in (i) the trend component being removed from the input window $\mathbf{x}$ before it is fed into the seasonality stack and (ii) the partial forecasts of trend and seasonality are available as separate interpretable outputs. Structurally, each of the stacks consists of several blocks connected with residual connections as depicted in Fig. 1 and each of them shares its respective, non-learnable $g_{s,\ell}^b$ and $g_{s,\ell}^f$. The number of blocks is 3 for both trend and seasonality. We found that on top of sharing $g_{s,\ell}^b$ and $g_{s,\ell}^f$, sharing all the weights across blocks in a stack resulted in better validation performance.

### 3.4 ENSEMBLING

Ensembling is used by all the top entries in the M4-competition. We rely on ensembling as well to be comparable. We found that ensembling is a much more powerful regularization technique than the popular alternatives, e.g. dropout or L2-norm penalty. The addition of those methods improved individual models, but was hurting the performance of the ensemble. The core property of an ensemble is diversity. We build an ensemble using several sources of diversity. First, the ensemble models are fit on three different metrics: sMAPE, MASE and MAPE, a version of sMAPE that has only the ground truth value in the denominator. Second, for every horizon $H$, individual models are trained on input windows of different length: $2H, 3H, \dots, 7H$, for a total of six window lengths. Thus the overall ensemble exhibits a multi-scale aspect. Finally, we perform a bagging procedure (Breiman, 1996) by including models trained with different random initializations. We use 180 total models to report results on the test set (please refer to Appendix B for the ablation of ensemble size). We use the median as ensemble aggregation function.

Table 1: Performance on the M4, M3, TOURISM test sets, aggregated over each dataset. Evaluation metrics are specified for each dataset; lower values are better. The number of time series in each dataset is provided in brackets.

| M4 Average (100,000) | | | M3 Average (3,003) | | TOURISM Average (1,311) | |
|---|---|---|---|---|---|---|
| | sMAPE | OWA | | sMAPE | | MAPE |
| Pure ML | 12.894 | 0.915 | Comb S-H-D | 13.52 | ETS | 20.88 |
| Statistical | 11.986 | 0.861 | ForecastPro | 13.19 | Theta | 20.88 |
| ProLogistica | 11.845 | 0.841 | Theta | 13.01 | ForePro | 19.84 |
| ML/TS combination | 11.720 | 0.838 | DOTM | 12.90 | Stratometrics | 19.52 |
| DL/TS hybrid | 11.374 | 0.821 | EXP | 12.71 | LeeCBaker | 19.35 |
| N-BEATS-G | 11.168 | 0.797 | | 12.47 | | **18.47** |
| N-BEATS-I | 11.174 | 0.798 | | 12.43 | | 18.97 |
| N-BEATS-I+G | **11.135** | **0.795** | | **12.37** | | 18.52 |

## 4 RELATED WORK

The approaches to TS forecasting can be split in a few distinct categories. The statistical modeling approaches based on exponential smoothing and its different flavors are well established and are often considered a default choice in the industry (Holt, 1957; 2004; Winters, 1960). More advanced variations of exponential smoothing include the winner of M3 competition, the Theta method (Assimakopoulos & Nikolopoulos, 2000) that decomposes the forecast into several theta-lines and statistically combines them. The pinnacle of the statistical approach encapsulates ARIMA, auto-ARIMA and in general, the unified state-space modeling approach, that can be used to explain and analyze all of the approaches mentioned above (see Hyndman & Khandakar (2008) for an overview). More recently, ML/TS combination approaches started infiltrating the domain with great success, showing promising results by using the outputs of statistical engines as features. In fact, 2 out of top-5 entries in the M4 competition are approaches of this type, including the second entry (Montero-Manso et al., 2019). The second entry computes the outputs of several statistical methods on the M4 dataset and combines them using gradient boosted tree (Chen & Guestrin, 2016). Somewhat independently, the work in the modern deep learning TS forecasting developed based on variations of recurrent neural networks (Flunkert et al., 2017; Rangapuram et al., 2018b; Toubeau et al., 2019; Zia & Razzaq, 2018) being largely dominated by the electricity load forecasting in the multi-variate setup. A few earlier works explored the combinations of recurrent neural networks with dilation, residual connections and attention (Chang et al., 2017; Kim et al., 2017; Qin et al., 2017). These served as a basis for the winner of the M4 competition (Smyl, 2020). The winning entry combines a Holt-Winters style seasonality model with its parameters fitted to a given TS via gradient descent and a unique combination of dilation/residual/attention approaches for each forecast horizon. The resulting model is a hybrid model that architecturally heavily relies on a time-series engine. It is hand crafted to each specific horizon of M4, making this approach hard to generalize to other datasets.

## 5 EXPERIMENTAL RESULTS

Our key empirical results based on aggregate performance metrics over several datasets—M4 (M4 Team, 2018b; Makridakis et al., 2018b), M3 (Makridakis & Hibon, 2000; Makridakis et al., 2018a) and TOURISM (Athanasopoulos et al., 2011)—appear in Table 1. More detailed descriptions of the datasets are provided in Section 5.1 and Appendix A. For each dataset, we compare our results with best 5 entries for this dataset reported in the literature, according to the customary metrics specific to each dataset (M4: OWA and sMAPE, M3: sMAPE, TOURISM: MAPE). More granular dataset-specific results with data splits over forecast horizons and types of time series appear in respective appendices (M4: Appendix C.1; M3: Appendix C.2; TOURISM: Appendix C.3).

In Table 1, we study the performance of two N-BEATS configurations: generic (N-BEATS-G) and interpretable (N-BEATS-I), as well as N-BEATS-I+G (ensemble of all models from N-BEATS-G and N-BEATS-I). **On M4 dataset**, we compare against 5 representatives from the M4 competition (Makri-

dakis et al., 2018b): each best in their respective model class. *Pure ML* is the submission by B. Trotta, the best entry among the 6 pure ML models. *Statistical* is the best pure statistical model by N.Z. Legaki and K. Koutsouri. *ML/TS combination* is the model by P. Montero-Manso, T. Talagala, R.J. Hyndman and G. Athanasopoulos, second best entry, gradient boosted tree over a few statistical time series models. ProLogistica is the third entry in M4 based on the weighted ensemble of statistical methods. Finally, *DL/TS hybrid* is the winner of M4 competition (Smyl, 2020). **On the M3 dataset**, we compare against the *Theta* method (Assimakopoulos & Nikolopoulos, 2000), the winner of M3; *DOTA*, a dynamically optimized Theta model (Fiorucci et al., 2016); *EXP*, the most resent statistical approach and the previous state-of-the-art on M3 (Spiliotis et al., 2019); as well as *ForecastPro*, an off-the-shelf forecasting software that is based on model selection between exponential smoothing, ARIMA and moving average (Athanasopoulos et al., 2011; Assimakopoulos & Nikolopoulos, 2000). **On the TOURISM dataset**, we compare against 3 statistical benchmarks (Athanasopoulos et al., 2011): *ETS*, exponential smoothing with cross-validated additive/multiplicative model; *Theta* method; *ForePro*, same as *ForecastPro* in M3; as well as top 2 entries from the TOURISM Kaggle competition (Athanasopoulos & Hyndman, 2011): *Stratometrics*, an unknown technique; *LeeCBaker* (Baker & Howard, 2011), a weighted combination of Naïve, linear trend model, and exponentially weighted least squares regression trend.

According to Table 1, N-BEATS demonstrates state-of-the-art performance on three challenging non-overlapping datasets containing time series from very different domains, sampling frequencies and seasonalities. As an example, on M4 dataset, the OWA gap between N-BEATS and the M4 winner $(0.821 - 0.795 = 0.026)$ is greater than the gap between the M4 winner and the second entry $(0.838 - 0.821 = 0.017)$. Generic N-BEATS model uses as little prior knowledge as possible, with no feature engineering, no scaling and no internal architectural components that may be considered TS-specific. Thus the result in Table 1 leads us to the conclusion that DL does not need support from the statistical approaches or hand-crafted feature engineering and domain knowledge to perform extremely well on a wide array of TS forecasting tasks. On top of that, the proposed general architecture performs very well on three different datasets outperforming a wide variety of models, both generic and manually crafted to respective dataset, including the winner of M4, a model architecturally adjusted by hand to each forecast-horizon subset of the M4 data.

## 5.1 DATASETS

**M4** (M4 Team, 2018b; Makridakis et al., 2018b) is the latest in an influential series of forecasting competitions organized by Spyros Makridakis since 1982 (Makridakis et al., 1982). The 100k-series dataset is large and diverse, consisting of data frequently encountered in business, financial and economic forecasting, and sampling frequencies ranging from hourly to yearly. A table with summary statistics is presented in Appendix A.1, showing wide variability in TS characteristics.

**M3** (Makridakis & Hibon, 2000) is similar in its composition to M4, but has a smaller overall scale (3003 time series total vs. 100k in M4). A table with summary statistics is presented in Appendix A.2. Over the past 20 years, this dataset has supported significant efforts in the design of more optimal statistical models, e.g. Theta and its variants (Assimakopoulos & Nikolopoulos, 2000; Fiorucci et al., 2016; Spiliotis et al., 2019). Furthermore, a recent publication (Makridakis et al., 2018a) based on a subset of M3 presented evidence that ML models are inferior to the classical statistical models.

**TOURISM** (Athanasopoulos et al., 2011) dataset was released as part of the respective Kaggle competition conducted by Athanasopoulos & Hyndman (2011). The data include monthly, quarterly and yearly series supplied by both governmental tourism organizations (e.g. Tourism Australia, the Hong Kong Tourism Board and Tourism New Zealand) as well as various academics, who had used them in previous studies. A table with summary statistics is presented in Appendix A.3.

## 5.2 TRAINING METHODOLOGY

We split each dataset into train, validation and test subsets. The test subset is the standard test set previously defined for each dataset (M4 Team, 2018a; Makridakis & Hibon, 2000; Athanasopoulos et al., 2011). The validation and train subsets for each dataset are obtained by splitting their full train sets at the boundary of the last horizon of each time series. We use the train and validation subsets to tune hyperparameters. Once the hyperparameters are determined, we train the model on the full train set and report results on the test set. Please refer to Appendix D for detailed hyperparameter settings

at the block level. N-BEATS is implemented and trained in Tensorflow (Abadi et al., 2015). We share parameters of the network across horizons, therefore we train one model per horizon for each dataset. If every time series is interpreted as a separate task, this can be linked back to the multitask learning and furthermore to meta-learning (see discussion in Section 6), in which a neural network is regularized by learning on multiple tasks to improve generalization. We would like to stress that models for different horizons and datasets reuse the same architecture. Architectural hyperparameters (width, number of layers, number of stacks, etc.) are fixed to the same values across horizons and across datasets (see Appendix D). The fact that we can reuse architecture and even hyperparameters across horizons indicates that the proposed architecture design generalizes well across time series of different nature. The same architecture is successfully trained on the M4 Monthly subset with 48k time series and the M3 Others subset with 174 time series. This is a much stronger result than *e.g.* the result of S. Smyl (Makridakis et al., 2018b) who had to use very different architectures hand crafted for different horizons.

To update network parameters for one horizon, we sample train batches of fixed size 1024. We pick 1024 TS ids from this horizon, uniformly at random with replacement. For each selected TS id we pick a random forecast point from the historical range of length $L_H$ immediately preceding the last point in the train part of the TS. $L_H$ is a cross-validated hyperparameter. We observed that for subsets with large number of time series it tends to be smaller and for subsets with smaller number of time series it tends to be larger. For example, in massive Yearly, Monthly, Quarterly subsets of M4 $L_H$ is equal to 1.5; and in moderate to small Weekly, Daily, Hourly subsets of M4 $L_H$ is equal to 10. Given a sampled forecast point, we set one horizon worth of points following it to be the target forecast window $\mathbf{y}$ and we set the history of points of one of lengths $2H, 3H, \ldots, 7H$ preceding it to be the input $\mathbf{x}$ to the network. We use the Adam optimizer with default settings and initial learning rate 0.001. While optimising the ensemble members relying on the minimization of sMAPE metric, we stop the gradient flows in the denominator to make training numerically stable. The neural network training is run with early stopping and the number of batches is determined on the validation set. The GPU based training of one ensemble member for entire M4 dataset takes between 30 min and 2 hours depending on neural network settings and hardware.

### 5.3 INTERPRETABILITY RESULTS

Fig. 2 studies the outputs of the proposed model in the generic and the interpretable configurations. As discussed in Section 3.3, to make the generic architecture presented in Fig. 1 interpretable, we constrain $g_\theta$ in the first stack to have the form of polynomial (2) while the second one has the form of Fourier basis (3). Furthermore, we use the outputs of the generic configuration of N-BEATS as control group (the generic model of 30 residual blocks depicted in Fig. 1 is divided into two stacks) and we plot both generic (suffix "-G") and interpretable (suffix "-I") stack outputs side by side in Fig. 2. The outputs of generic model are arbitrary and non-interpretable: either trend or seasonality or both of them are present at the output of both stacks. The magnitude of the output (peak-to-peak) is generally smaller at the output of the second stack. The outputs of the interpretable model exhibit distinct properties: the trend output is monotonic and slowly moving, the seasonality output is regular, cyclical and has recurring fluctuations. The peak-to-peak magnitude of the seasonality output is significantly larger than that of the trend, if significant seasonality is present in the time series. Similarly, the peak-to-peak magnitude of trend output tends to be small when no obvious trend is present in the ground truth signal. Thus the proposed interpretable architecture decomposes its forecast into two distinct components. Our conclusion is that the outputs of the DL model can be made interpretable by encoding a sensible inductive bias in the architecture. Table 1 confirms that this does not result in performance drop.

## 6 DISCUSSION: CONNECTIONS TO META-LEARNING

Meta-learning defines an inner *learning procedure* and an outer *learning procedure*. The inner learning procedure is parameterized, conditioned or otherwise influenced by the outer learning procedure (Bengio et al., 1991). The prototypical inner vs. outer learning is individual learning in the lifetime of an animal vs. evolution of the inner learning procedure itself over many generations of individuals. To see the two levels, it often helps to refer to two sets of parameters, the inner parameters (e.g. synaptic weights) which are modified inside the inner learning procedure, and the

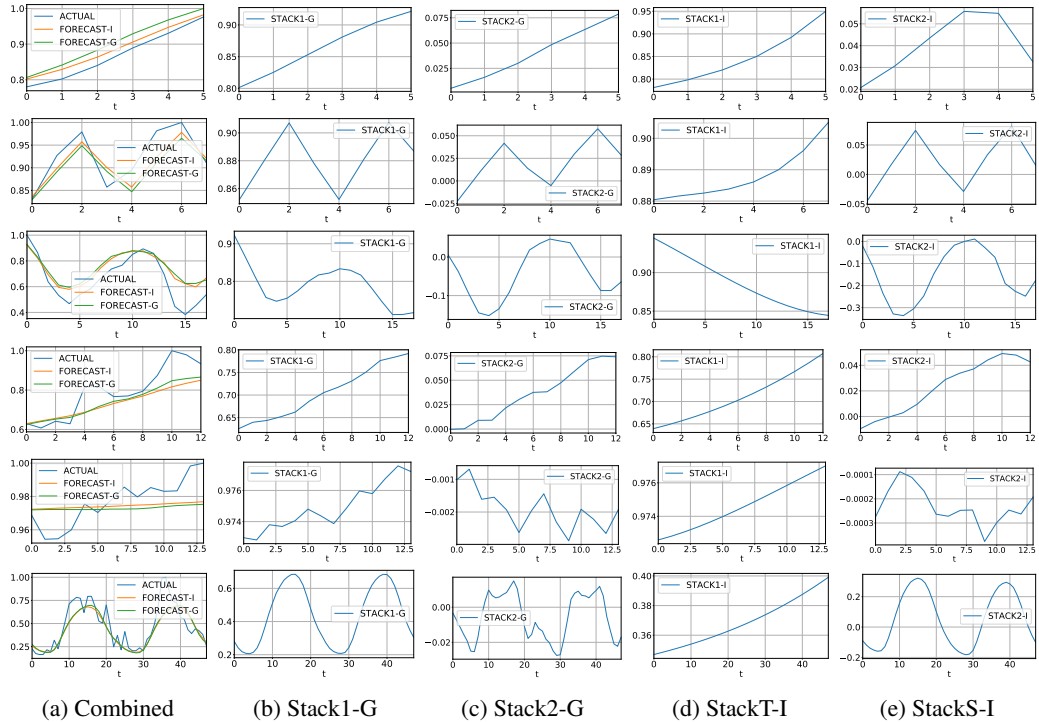

|          |          |          |          |          |
|:--------:|:--------:|:--------:|:--------:|:--------:|
| (a) Combined | (b) Stack1-G | (c) Stack2-G | (d) StackT-I | (e) StackS-I |

Figure 2: The outputs of generic and the interpretable configurations, M4 dataset. Each row is one time series example per data frequency, top to bottom (Yearly: id Y3974, Quarterly: id Q11588, Monthly: id M19006, Weekly: id W246, Daily: id D404, Hourly: id H344). The magnitudes in a row are normalized by the maximal value of the actual time series for convenience. Column (a) shows the actual values (ACTUAL), the generic model forecast (FORECAST-G) and the interpretable model forecast (FORECAST-I). Columns (b) and (c) show the outputs of stacks 1 and 2 of the generic model, respectively; FORECAST-G is their summation. Columns (d) and (e) show the output of the Trend and the Seasonality stacks of the interpretable model, respectively; FORECAST-I is their summation.

outer parameters or meta-parameters (e.g. genes) which get modified only in the outer learning procedure.

N-BEATS can be cast as an instance of meta-learning by drawing the following parallels. The outer learning procedure is encapsulated in the parameters of the whole network, learned by gradient descent. The inner learning procedure is encapsulated in the set of basic building blocks and modifies the expansion coefficients $\theta^f$ that basis $g^f$ takes as inputs. The inner learning proceeds through a sequence of stages, each corresponding to a block within the stack of the architecture. Each of the blocks can be thought of as performing the equivalent of an update step which gradually modifies the expansion coefficients $\theta^f$ which eventually feed into $g^f$ in each block (which get added together to form the final prediction). The inner learning procedure takes a single history from a piece of a TS and sees that history as a training set. It produces forward expansion coefficients $\theta^f$ (see Fig. 1), which parametrically map inputs to predictions. In addition, each preceding block modifies the input to the next block by producing backward expansion coefficients $\theta^b$, thus conditioning the learning and the output of the next block. In the case of the interpretable model, the meta-parameters are only in the FC layers because the $g^f$'s are fixed. In the case of the generic model, the meta-parameters also include the $\mathbf{V}$'s which define the $g^f$ non-parametrically. This point of view is further reinforced by the results of the ablation study reported in Appendix B showing that increasing the number of blocks in the stack, as well as the number of stacks improves generalization performance, and can be interpreted as more iterations of the inner learning procedure.

## 7 CONCLUSIONS

We proposed and empirically validated a novel architecture for univariate TS forecasting. We showed that the architecture is general, flexible and it performs well on a wide array of TS forecasting problems. We applied it to three non-overlapping challenging competition datasets: M4, M3 and TOURISM and demonstrated state-of-the-art performance in two configurations: generic and interpretable. This allowed us to validate two important hypotheses: (i) the generic DL approach performs exceptionally well on heterogeneous univariate TS forecasting problems using no TS domain knowledge, (ii) it is viable to additionally constrain a DL model to force it to decompose its forecast into distinct human interpretable outputs. We also demonstrated that the DL models can be trained on multiple time series in a multi-task fashion, successfully transferring and sharing individual learnings. We speculate that N-BEATS's performance can be attributed in part to it carrying out a form of meta-learning, a deeper investigation of which should be the subject of future work.

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

Table 2: Composition of the M4 dataset: the number of time series based on their sampling frequency and type.

| Type | Frequency / Horizon | | | | | | Total |
|---|---|---|---|---|---|---|---|
| | Yearly/6 | Qtly/8 | Monthly/18 | Wkly/13 | Daily/14 | Hrly/48 | |
| Demographic | 1,088 | 1,858 | 5,728 | 24 | 10 | 0 | 8,708 |
| Finance | 6,519 | 5,305 | 10,987 | 164 | 1,559 | 0 | 24,534 |
| Industry | 3,716 | 4,637 | 10,017 | 6 | 422 | 0 | 18,798 |
| Macro | 3,903 | 5,315 | 10,016 | 41 | 127 | 0 | 19,402 |
| Micro | 6,538 | 6,020 | 10,975 | 112 | 1,476 | 0 | 25,121 |
| Other | 1,236 | 865 | 277 | 12 | 633 | 414 | 3,437 |
| Total | 23,000 | 24,000 | 48,000 | 359 | 4,227 | 414 | 100,000 |
| Min. Length | 19 | 24 | 60 | 93 | 107 | 748 | |
| Max. Length | 841 | 874 | 2812 | 2610 | 9933 | 1008 | |
| Mean Length | 37.3 | 100.2 | 234.3 | 1035.0 | 2371.4 | 901.9 | |
| SD Length | 24.5 | 51.1 | 137.4 | 707.1 | 1756.6 | 127.9 | |
| % Smooth | 82% | 89% | 94% | 84% | 98% | 83% | |
| % Erratic | 18% | 11% | 6% | 16% | 2% | 17% | |

## A   DATASET DETAILS

### A.1   M4 DATASET DETAILS

Table 2 outlines the composition of the M4 dataset across domains and forecast horizons by listing the number of time series based on their frequency and type (M4 Team, 2018b). The M4 dataset is large and diverse: all forecast horizons are composed of heterogeneous time series types (with exception of Hourly) frequently encountered in business, financial and economic forecasting. Summary statistics on series lengths are also listed, showing wide variability therein, as well as a characterization (*smooth* vs *erratic*) that follows Syntetos et al. (2005), and is based on the squared coefficient of variation of the series. All series have positive observed values at all time-steps; as such, none can be considered *intermittent* or *lumpy* per Syntetos et al. (2005).

### A.2   M3 DATASET DETAILS

Table 3 outlines the composition of the M3 dataset across domains and forecast horizons by listing the number of time series based on their frequency and type (Makridakis & Hibon, 2000). The M3 is smaller than the M4, but it is still large and diverse: all forecast horizons are composed of heterogeneous time series types frequently encountered in business, financial and economic forecasting. Summary statistics on series lengths are also listed, showing wide variability in length, as well as a characterization (*smooth* vs *erratic*) that follows Syntetos et al. (2005), and is based on the squared coefficient of variation of the series. All series have positive observed values at all time-steps; as such, none can be considered *intermittent* or *lumpy* per Syntetos et al. (2005).

### A.3   TOURISM DATASET DETAILS

Table 4 outlines the composition of the TOURISM dataset across forecast horizons by listing the number of time series based on their frequency. Summary statistics on series lengths are listed, showing wide variability in length. All series have positive observed values at all time-steps. In contrast to M4 and M3 datasets, TOURISM includes a much higher fraction of erratic series.

Table 3: Composition of the M3 dataset: the number of time series based on their sampling frequency and type.

| Type | Frequency / Horizon | | | | Total |
|---|---|---|---|---|---|
| | Yearly/6 | Quarterly/8 | Monthly/18 | Other/8 | |
| Demographic | 245 | 57 | 111 | 0 | 413 |
| Finance | 58 | 76 | 145 | 29 | 308 |
| Industry | 102 | 83 | 334 | 0 | 519 |
| Macro | 83 | 336 | 312 | 0 | 731 |
| Micro | 146 | 204 | 474 | 4 | 828 |
| Other | 11 | 0 | 52 | 141 | 204 |
| Total | 645 | 756 | 1,428 | 174 | 3,003 |
| Min. Length | 20 | 24 | 66 | 71 | |
| Max. Length | 47 | 72 | 144 | 104 | |
| Mean Length | 28.4 | 48.9 | 117.3 | 76.6 | |
| SD Length | 9.9 | 10.6 | 28.5 | 10.9 | |
| % Smooth | 90% | 99% | 98% | 100% | |
| % Erratic | 10% | 1% | 2% | 0% | |

Table 4: Composition of the TOURISM dataset: the number of time series based on their sampling frequency.

| | Frequency / Horizon | | | Total |
|---|---|---|---|---|
| | Yearly/4 | Quarterly/8 | Monthly/24 | |
| | 518 | 427 | 366 | 1,311 |
| Min. Length | 11 | 30 | 91 | |
| Max. Length | 47 | 130 | 333 | |
| Mean Length | 24.4 | 99.6 | 298 | |
| SD Length | 5.5 | 20.3 | 55.7 | |
| % Smooth | 77% | 61% | 49% | |
| % Erratic | 23% | 39% | 51% | |

Table 5: sMAPE on the validation set, generic architecture. sMAPE for varying number of stacks, each having one residual block.

| Stacks | sMAPE |
|--------|--------|
| 1 | 11.154 |
| 3 | 11.061 |
| 9 | 10.998 |
| 18 | 10.950 |
| 30 | 10.937 |

Table 6: sMAPE on the validation set, interpretable architecture. Ablation of the synergy of the layers with different basis functions and multi-block stack gain.

| Detrend | Seasonality | sMAPE |
|---------|-------------|--------|
| 0 | 2 | 11.189 |
| 2 | 0 | 11.572 |
| 1 | 1 | 11.040 |
| 3 | 3 | 10.986 |

## B ABLATION STUDIES

### B.1 LAYER STACKING AND BASIS SYNERGY

We performed an ablation study on the validation set, using sMAPE metric as performance criterion. We addressed two specific questions with this study. First, Is stacking layers helpful? Second, Does the architecture based on the combination of layers with different basis functions results in better performance than the architecture using only one layer type?

**Layer stacking.** We start our study with the generic architecture that consists of stacks of one residual block of 5 FC layers each of the form Fig. 1 and we increase the number of stacks. Results presented in Table 5 confirm that increasing the number of stacks decreases error and at certain point the gain saturates. We would like to mention that the network having 30 stack of depth 5 is in fact a very deep network of total depth 150 layers.

**Basis synergy.** Stacking works well for the interpretable architecture as can be seen in Table 6 depicting the results of ablating the interpretable architecture configuration. Here we experiment with the architecture that is composed of 2 stacks, stack one is trend model and stack two is the seasonality model. Each stack has variable number of residual blocks and each residual block has 5 FC layers. We found that this architecture works best when all weights are shared within stack. We clearly see that increasing the number of layers improves performance. The largest network is 60 layers deep. On top of that, we observe that the architecture that consists of stacks based on different basis functions wins over the architecture based on the same stack. It looks like chaining stacks of different nature results in synergistic effects. This is logical as function classes that can be modelled by trend and seasonality stacks have small overlap.

### B.2 ENSEMBLE SIZE

Figure 3 demonstrates that increasing the ensemble size results in improved performance. Most importantly, according to Figure 3, N-BEATS achieves state-of-the-art performance even if comparatively small ensemble size of 18 models is used. Therefore, computational efficiency of N-BEATS can be traded very effectively for performance and there is no over-reliance of the results on large ensemble size.

### B.3 DOUBLY RESIDUAL STACKING

In Section 3.2 we described the proposed doubly residual stacking (DRESS) principle, which is the topological foundation of N-BEATS. The topology is based on both (i) running a residual backcast connection and (ii) producing partial block-level forecasts that are further aggregated at stack and model levels to produce the final model-level forecast. In this section we conduct a study to confirm the accuracy effectiveness of this topology compared to several alternatives. The methodology underlying this study is that we remove either the backcast or partial forecast links or both and track how this affects the forecasting metrics. We keep the number of parameters in the network for each of the architectural alternatives fixed by using the same number of layers in the network (we used default hyperparameter settings reported in Table 18). The architectural alternatives are depicted in Figure 4 and described in detail below.

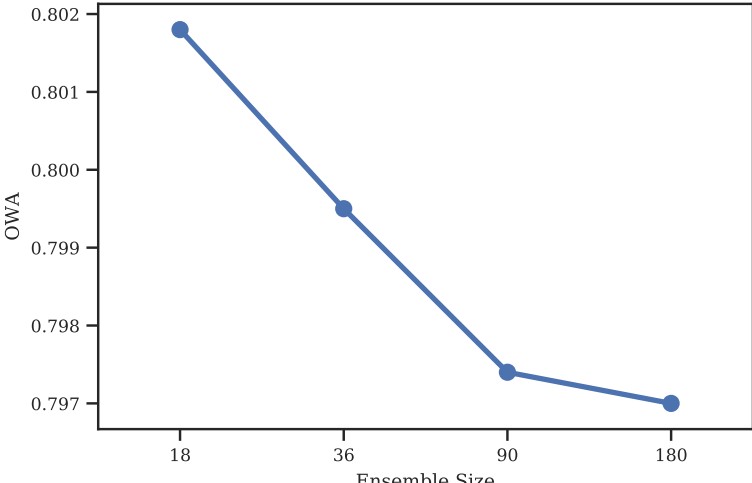

Figure 3: M4 test performance (OWA) as a function of ensemble size, based on N-BEATS-G. This figure shows that N-BEATS loses less than 0.5% in terms of OWA performance even if 10 times smaller ensemble size is used.

**N-BEATS-DRESS** is depicted in Fig. 4a. This is the default configuration of N-BEATS using doubly residual stacking described in Section 3.2.

**PARALLEL** is depicted in Fig. 4b. This is the alternative where the backward residual connection is disabled and the overall model input is fed to every block. The blocks then forecast in parallel using the same input and their individual outputs are summed to make the final forecast.

**NO-RESIDUAL** is depicted in Fig. 4c. This is the alternative where the backward residual connection is disabled. Unlike PARALLEL, in this case the backcast forecast of the previous block is fed as input to the next block. Unlike the usual feed-forward network, in the NO-RESIDUAL architecture, each block makes a partial forecast and their individual outputs are summed to make the final forecast.

**LAST-FORWARD** is depicted in Fig. 4d. This is the alternative where the backward residual connection is active, however the model level forecast is derived only from the last block. So, the partial forward forecasts are disabled. This is the architecture that is closest to the classical residual network.

**NO-RESIDUAL-LAST-FORWARD** is depicted in Fig. 4f. This is the alternative where both backward residual and the partial forward connections are disabled. This is therefore a simple feed-forward network, but very deep.

The quantitative ablation study results on the M4 dataset are reported in Tables 7–10. N-BEATS-DRESS model is essentially N-BEATS model in this study. For this study we used ensemble size of 18. Since the ensemble size is 18 for N-BEATS-DRESS, as opposed to 180 used for N-BEATS, the OWA metric reported in Table 9 for N-BEATS-DRESS is higher than the OWA reported for N-BEATS-G in Table 12. Note that both results align well with OWA reported in Figure 3 for different ensemble sizes, as part of the ensemble size ablation conducted in Section B.2.

The results presented in Tables 7–10 demonstrate that the doubly residual stacking topology provides a clear overall advantage over the alternative architectures in which either backcast residual links or the partial forward forecast links are disabled.

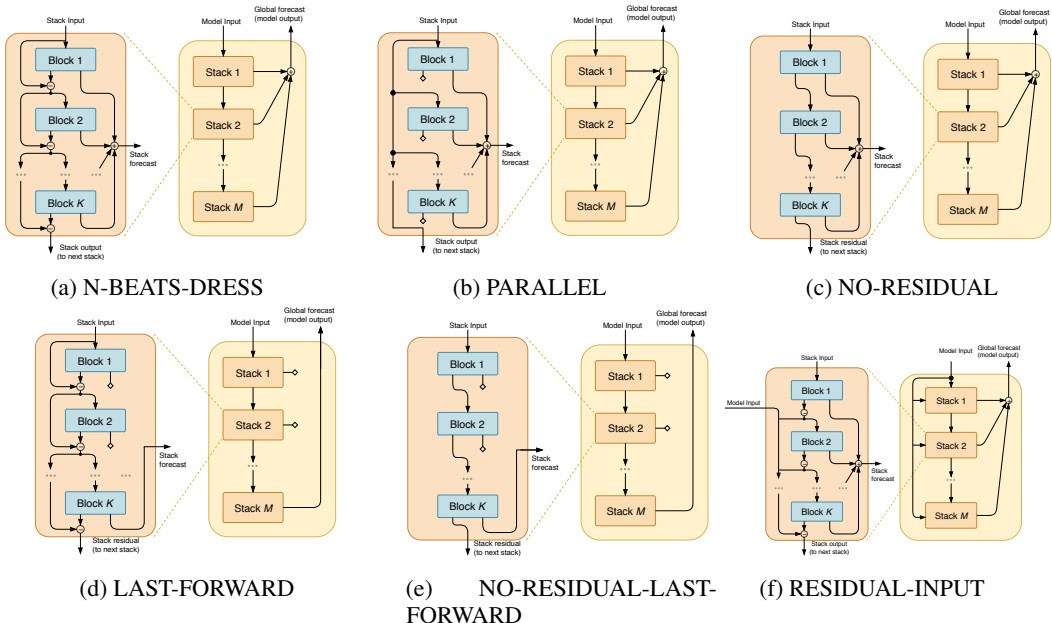

(a) N-BEATS-DRESS  (b) PARALLEL  (c) NO-RESIDUAL

(d) LAST-FORWARD  (e)  NO-RESIDUAL-LAST-FORWARD  (f) RESIDUAL-INPUT

Figure 4: The architectural configurations used in the ablation study of the doubly residual stack. Symbol ⋄ denotes unconnected output.

Table 7: Performance on the M4 test set, sMAPE. Lower values are better. The results are obtained on the ensemble of 18 generic models.

|  | Yearly (23k) | Quarterly (24k) | Monthly (48k) | Others (5k) | Average (100k) |
|---|---|---|---|---|---|
| PARALLEL-G | 13.279 | 9.558 | 12.510 | 3.691 | 11.538 |
| NO-RESIDUAL-G | 13.195 | 9.555 | 12.451 | 3.759 | 11.493 |
| LAST-FORWARD-G | 13.200 | 9.322 | 12.352 | 3.703 | 11.387 |
| NO-RESIDUAL-LAST-FORWARD-G | 15.386 | 11.346 | 15.282 | 6.673 | 13.931 |
| RESIDUAL-INPUT-G | 13.264 | 9.545 | 12.316 | 3.692 | 11.438 |
| N-BEATS-DRESS-G | 13.211 | 9.217 | 12.122 | 3.636 | 11.251 |

Table 8: Performance on the M4 test set, sMAPE. Lower values are better. The results are obtained on the ensemble of 18 interpretable models.

|  | Yearly (23k) | Quarterly (24k) | Monthly (48k) | Others (5k) | Average (100k) |
|---|---|---|---|---|---|
| PARALLEL-I | 13.207 | 9.530 | 12.500 | 3.710 | 11.510 |
| NO-RESIDUAL-I | 13.075 | 9.707 | 12.708 | 4.007 | 11.637 |
| LAST-FORWARD-I | 13.168 | 9.547 | 12.111 | 3.599 | 11.313 |
| NO-RESIDUAL-LAST-FORWARD-I | 13.067 | 10.207 | 15.177 | 4.912 | 12.986 |
| RESIDUAL-INPUT-I | 13.104 | 9.716 | 12.814 | 4.005 | 11.697 |
| N-BEATS-DRESS-I | 13.155 | 9.286 | 12.009 | 3.642 | 11.201 |

Table 9: Performance on the M4 test set, OWA. Lower values are better. The results are obtained on the ensemble of 18 generic models.

| | Yearly (23k) | Quarterly (24k) | Monthly (48k) | Others (5k) | Average (100k) |
|---|---|---|---|---|---|
| PARALLEL-G | 0.780 | 0.832 | 0.852 | 0.844 | 0.822 |
| NO-RESIDUAL-G | 0.774 | 0.831 | 0.851 | 0.853 | 0.819 |
| LAST-FORWARD-G | 0.774 | 0.808 | 0.840 | 0.846 | 0.811 |
| NO-RESIDUAL-LAST-FORWARD-G | 0.948 | 1.029 | 1.095 | 1.296 | 1.030 |
| RESIDUAL-INPUT-G | 0.779 | 0.831 | 0.840 | 0.844 | 0.817 |
| N-BEATS-DRESS-G | 0.776 | 0.800 | 0.823 | 0.835 | 0.803 |

Table 10: Performance on the M4 test set, OWA. Lower values are better. The results are obtained on the ensemble of 18 interpretable models.

| | Yearly (23k) | Quarterly (24k) | Monthly (48k) | Others (5k) | Average (100k) |
|---|---|---|---|---|---|
| PARALLEL-I | 0.776 | 0.831 | 0.857 | 0.845 | 0.821 |
| NO-RESIDUAL-I | 0.769 | 0.848 | 0.886 | 0.886 | 0.833 |
| LAST-FORWARD-I | 0.773 | 0.836 | 0.825 | 0.817 | 0.808 |
| NO-RESIDUAL-LAST-FORWARD-I | 0.771 | 0.900 | 1.085 | 1.016 | 0.922 |
| RESIDUAL-INPUT-I | 0.771 | 0.848 | 0.892 | 0.887 | 0.836 |
| N-BEATS-DRESS-I | 0.771 | 0.805 | 0.819 | 0.836 | 0.800 |

Table 11: Performance on the M4 test set, sMAPE. Lower values are better. Red – second best.

|  | Yearly (23k) | Quarterly (24k) | Monthly (48k) | Others (5k) | Average (100k) |
|---|---|---|---|---|---|
| Best pure ML | 14.397 | 11.031 | 13.973 | 4.566 | 12.894 |
| Best statistical | 13.366 | 10.155 | 13.002 | 4.682 | 11.986 |
| Best ML/TS combination | 13.528 | 9.733 | 12.639 | 4.118 | 11.720 |
| DL/TS hybrid, M4 winner | 13.176 | 9.679 | 12.126 | 4.014 | 11.374 |
| N-BEATS-G | 13.023 | **9.212** | 12.048 | **3.574** | 11.168 |
| N-BEATS-I | 12.924 | 9.287 | 12.059 | 3.684 | 11.174 |
| N-BEATS-I+G | **12.913** | 9.213 | **12.024** | 3.643 | **11.135** |

Table 12: Performance on the M4 test set, OWA and M4 rank. Lower values are better. Red – second best.

|  | Yearly (23k) | Quarterly (24k) | Monthly (48k) | Others (5k) | Average (100k) | Rank |
|---|---|---|---|---|---|---|
| Best pure ML | 0.859 | 0.939 | 0.941 | 0.991 | 0.915 | 23 |
| Best statistical | 0.788 | 0.898 | 0.905 | 0.989 | 0.861 | 8 |
| Best ML/TS combination | 0.799 | 0.847 | 0.858 | 0.914 | 0.838 | 2 |
| DL/TS hybrid, M4 winner | 0.778 | 0.847 | 0.836 | 0.920 | 0.821 | 1 |
| N-BEATS-G | 0.765 | **0.800** | 0.820 | **0.822** | 0.797 |  |
| N-BEATS-I | **0.758** | 0.807 | 0.824 | 0.849 | 0.798 |  |
| N-BEATS-I+G | **0.758** | **0.800** | **0.819** | 0.840 | **0.795** |  |

## C    DETAILED EMPIRICAL RESULTS

### C.1    DETAILED RESULTS: M4 DATASET

Tables 11 and 12 present our key quantitative empirical results showing that the proposed model achieves the state of the art performance on the challenging M4 benchmark. We study the performance of two model configurations: generic (Ours-G) and interpretable (Ours-I), as well as Ours-I+G (ensemble of all models from Ours-G and Ours-I). We compare against 4 representatives from the M4 competition: each best in their respective model class. *Best pure ML* is the submission by B. Trotta, the best entry among the 6 pure ML models. *Best statistical* is the best pure statistical model by N.Z. Legaki and K. Koutsouri. *Best ML/TS combination* is the model by P. Montero-Manso, T. Talagala, R.J. Hyndman and G. Athanasopoulos, second best entry, gradient boosted tree over a few statistical time series models. Finally, *DL/TS hybrid* is the winner of M4 competition (Smyl, 2020).

N-BEATS outperforms all other approaches on all the studied subsets of time series. The average OWA gap between our generic model and the M4 winner ($0.821 - 0.795 = 0.026$) is greater than the gap between the M4 winner and the second entry ($0.838 - 0.821 = 0.017$).

A more granular and detailed statistical analysis of our results on M4 is provided in Table 13. This table first presents the sMAPE for N-BEATS, decomposed by M4 time series sub-type and sampling frequency (upper part). Then (lower part), it shows the *average* sMAPE *difference* between the N-BEATS results and the M4 winner (TS/DL hybrid by S. Smyl), adding the standard error of that difference (in parentheses); bold entries indicate statistical significance at the 99% level based on a two-sided paired $t$-test.

We note that each cross-section of the M4 dataset into horizon and type may be regarded as an independent mini-dataset. We observe that over those mini-datasets there is a preponderance of statistically significant differences between N-BEATS and Smyl (18 cases out of 31) to the advantage of N-BEATS. This provides evidence that (i) the improvement observed on average in Tables 11 and 12 is statistically significant and consistent over smaller subsets of M4 and (ii) N-BEATS generalizes well over time series of different types and sampling frequencies.

Table 13: Performance decomposition on non-overlapping subsets of the M4 test set and comparison with the Smyl model results.

|  | Demographic | Finance | Industry | Macro | Micro | Other |
|---|---|---|---|---|---|---|
| **sMAPE per M4 series type and sampling frequency** | | | | | | |
| Yearly | 8.931 | 13.741 | 16.317 | 13.327 | 10.489 | 13.320 |
| Quarterly | 9.219 | 10.787 | 8.628 | 8.576 | 9.264 | 6.250 |
| Monthly | 4.357 | 13.353 | 12.657 | 12.571 | 13.627 | 11.595 |
| Weekly | 4.580 | 3.004 | 9.258 | 7.220 | 10.425 | 6.183 |
| Daily | 6.351 | 3.467 | 3.835 | 2.525 | 2.299 | 2.885 |
| Hourly | | | | | | 8.197 |
| **Average sMAPE difference vs Smyl model**, computed as N-BEATS − Smyl. *Standard error of the mean displayed in parenthesis.* *Bold entries are significant at the 99% level (2-sided paired t-test).* | | | | | | |
| Yearly | **−0.749** | **−0.337** | −0.065 | **−0.386** | **−0.168** | −0.157 |
|  | (**0.119**) | (0.065) | (0.087) | (**0.085**) | (**0.056**) | (0.140) |
| Quarterly | **−0.651** | **−0.281** | **−0.328** | **−0.712** | **−0.523** | −0.029 |
|  | (**0.085**) | (**0.047**) | (**0.043**) | (**0.060**) | (**0.051**) | (0.083) |
| Monthly | **−0.185** | **−0.379** | **−0.419** | 0.089 | **0.338** | −0.279 |
|  | (**0.023**) | (**0.034**) | (**0.036**) | (0.039) | (**0.034**) | (0.162) |
| Weekly | −0.336 | **−1.075** | −0.937 | −1.627 | **−3.029** | −1.193 |
|  | (0.270) | (**0.221**) | (1.399) | (0.770) | (**0.378**) | (0.772) |
| Daily | 0.191 | **−0.098** | **−0.124** | −0.026 | **−0.367** | −0.037 |
|  | (0.231) | (**0.018**) | (**0.025**) | (0.057) | (**0.013**) | (0.015) |
| Hourly | | | | | | **−1.132** |
|  | | | | | | (**0.163**) |

Table 14: Performance on the M3 test set, Average SMAPE, aggregate over all forecast horizons (Yearly: 1-6, Quarterly: 1-8, Monthly: 1-18, Other: 1-8, Average: 1-18). Lower values are better. Red – second best. [†]Numbers are computed by us.

|  | Yearly (645) | Quarterly (756) | Monthly (1428) | Others (174) | Average (3003) |
|---|---|---|---|---|---|
| Naïve2 | 17.88 | 9.95 | 16.91 | 6.30 | 15.47 |
| ARIMA (B–J automatic) | 17.73 | 10.26 | 14.81 | 5.06 | 14.01 |
| Comb S-H-D | 17.07 | 9.22 | 14.48 | 4.56 | 13.52 |
| ForecastPro | 17.14 | 9.77 | 13.86 | 4.60 | 13.19 |
| Theta | 16.90 | 8.96 | 13.85 | 4.41 | 13.01 |
| DOTM (Fiorucci et al., 2016) | 15.94 | 9.28 | 13.74 | 4.58 | 12.90 |
| EXP (Spiliotis et al., 2019) | 16.39 | 8.98 | 13.43 | 5.46 | 12.71[†] |
| LGT (Smyl & Kuber, 2016) | **15.23** | n/a | n/a | 4.26 | n/a |
| BaggedETS.BC (Bergmeir et al., 2016) | 17.49 | 9.89 | 13.74 | n/a | n/a |
| N-BEATS-G | 16.2 | 8.92 | 13.19 | **4.19** | 12.47 |
| N-BEATS-I | 15.84 | 9.03 | 13.15 | 4.30 | 12.43 |
| N-BEATS-I+G | 15.93 | **8.84** | **13.11** | 4.24 | **12.37** |

## C.2 DETAILED RESULTS: M3 DATASET

Results for M3 dataset are provided in Table 14. The performance metric is calculated using the earlier version of SMAPE, defined specifically for the M3 competition:[1]

$$\text{SMAPE} = \frac{200}{H} \sum_{i=1}^{H} \frac{|y_{T+i} - \widehat{y}_{T+i}|}{y_{T+i} + \widehat{y}_{T+i}}. \tag{4}$$

For some of the methods, either average SMAPE was not reported or SMAPE for some of the splits was not reported in their respective publications. Below, we list those cases. BaggedETS.BC (Bergmeir et al., 2016) has not reported numbers on Others. LGT (Smyl & Kuber, 2016) did not report results on Monthly and Quarterly data. According to the authors, the underlying RNN had problems dealing with raw seasonal data, the ETS based pre-processing was not effective and the LGT pre-processing was not computationally feasible given comparatively large number of time series and their comparatively large length (Smyl & Kuber, 2016). Finally, EXP (Spiliotis et al., 2019) reported average performance computed using a different methodology than the default M3 and M4 methodology (source: personal communication with the authors). For the latter method we recomputed the Average SMAPE based on the previously reported Yearly, Quarterly and Monthly splits. To calculate it, we follow the M3, M4 and TOURISM competition methodology and compute the average metric as the average over all time series and over all forecast horizons. Given the performance metric values aggregated over Yearly, Quarterly and Monthly splits, the average can be computed straightforwardly as:

$$\text{SMAPE}_{\text{Average}} = \frac{N_{\text{Year}}}{N_{\text{Tot}}} \text{SMAPE}_{\text{Year}} + \frac{N_{\text{Quart}}}{N_{\text{Tot}}} \text{SMAPE}_{\text{Quart}} + \frac{N_{\text{Month}}}{N_{\text{Tot}}} \text{SMAPE}_{\text{Month}} + \frac{N_{\text{Others}}}{N_{\text{Tot}}} \text{SMAPE}_{\text{Others}}. \tag{5}$$

Here $N_{\text{Tot}} = N_{\text{Year}} + N_{\text{Quart}} + N_{\text{Month}} + N_{\text{Others}}$ and $N_{\text{Year}} = 6 \times 645, N_{\text{Quart}} = 8 \times 756, N_{\text{Month}} = 18 \times 1428, N_{\text{Others}} = 8 \times 174$. It is clear that for each split, its $N$ is the product of its respective number of time series and its largest forecast horizon.

---

[1]With minor differences compared to the SMAPE definition used for M4. Please refer to Appendix A in (Makridakis & Hibon, 2000) for the mathematical definition.

Table 15: Performance on the TOURISM test set, Average MAPE, aggregate over all forecast horizons (Yearly: 1-4, Quarterly: 1-8, Monthly: 1-24, Average: 1-24). Lower values are better. Red – second best.

| | Yearly (518) | Quarterly (427) | Monthly (366) | Average (1311) |
|---|---|---|---|---|
| **Statistical benchmarks** (Athanasopoulos et al., 2011) | | | | |
| SNaïve | 23.61 | 16.46 | 22.56 | 21.25 |
| Theta | 23.45 | 16.15 | 22.11 | 20.88 |
| ForePro | 26.36 | 15.72 | 19.91 | 19.84 |
| ETS | 27.68 | 16.05 | 21.15 | 20.88 |
| Damped | 28.15 | 15.56 | 23.47 | 22.26 |
| ARIMA | 28.03 | 16.23 | 21.13 | 20.96 |
| **Kaggle competitors** (Athanasopoulos & Hyndman, 2011) | | | | |
| SaliMali | n/a | 14.83 | 19.64 | n/a |
| LeeCBaker | 22.73 | 15.14 | 20.19 | 19.35 |
| Stratometrics | 23.15 | 15.14 | 20.37 | 19.52 |
| Robert | n/a | 14.96 | 20.28 | n/a |
| Idalgo | n/a | 15.07 | 20.55 | n/a |
| N-BEATS-G (Ours) | 21.67 | **14.71** | **19.17** | **18.47** |
| N-BEATS-I (Ours) | 21.55 | 15.22 | 19.82 | 18.97 |
| N-BEATS-I+G (Ours) | **21.44** | 14.78 | 19.29 | 18.52 |

## C.3 Detailed results: tourism Dataset

Detailed results for the TOURISM competition dataset are provided in Table 15. The respective Kaggle competition was divided into two parts: (i) Yearly time series forecasting and (ii) Quarterly/Monthly time series forecasting (Athanasopoulos & Hyndman, 2011). Some of the participants chose to take part only in the second part. Therefore, In addition to entries present in Table 1, we report competitors from (Athanasopoulos & Hyndman, 2011) that have missing results in Yearly competition. In particular, *SaliMali* team is the winner of the Quarterly/Monthly time series forecasting competition (Brierley, 2011). Their approach is based on a weighted ensemble of statistical methods. Teams *Robert* and *Idalgo* used unknown approaches. We can see from Table 15 that N-BEATS achieves state-of-the-art performance on all subsets of TOURISM dataset. On average, it is state of the art and it gains 4.2% over the best-known approach *LeeCBaker*, and 11.5% over auto-ARIMA.

The average metrics have not been reported in the original competition results (Athanasopoulos et al., 2011; Athanasopoulos & Hyndman, 2011). Therefore, in Table 15, we present the Average MAPE metric calculated by us based on the previously reported Yearly, Quarterly and Monthly splits. To calculate it, we follow the M4 competition methodology and compute the average metric as the average over all time series and over all forecast horizons. Given the performance metric values aggregated over Yearly, Quarterly and Monthly splits, the average can be computed straightforwardly as:

$$\text{MAPE}_{\text{Average}} = \frac{N_{\text{Year}}}{N_{\text{Tot}}} \text{MAPE}_{\text{Year}} + \frac{N_{\text{Quart}}}{N_{\text{Tot}}} \text{MAPE}_{\text{Quart}} + \frac{N_{\text{Month}}}{N_{\text{Tot}}} \text{MAPE}_{\text{Month}}. \tag{6}$$

Here $N_{\text{Tot}} = N_{\text{Year}} + N_{\text{Quart}} + N_{\text{Month}}$ and $N_{\text{Year}} = 4 \times 518, N_{\text{Quart}} = 8 \times 427, N_{\text{Month}} = 24 \times 366$. It is clear that for each split, its $N$ is the product of its respective number of time series and its largest forecast horizon.

## C.4 DETAILED RESULTS: ELECTRICITY AND TRAFFIC DATASETS

In this experiment we are comparing the performances of MatFact (Yu et al., 2016), DeepAR (Flunkert et al., 2017) (Amazon Labs), Deep State (Rangapuram et al., 2018a) (Amazon Labs), Deep Factors (Wang et al., 2019) (Amazon Labs), and N-BEATS models on ELECTRICITY[2] (Dua & Graff, 2017) and TRAFFIC[3] (Dua & Graff, 2017) datasets. The results are presented in in Table 16.

Both datasets are aggregated to hourly data, but using different aggregation operations: sum for ELECTRICITY and mean for TRAFFIC. The hourly aggregation is done so that all the points available in $(h-1:00, h:00]$ hours are aggregated to hour $h$, thus if original dataset starts on 2011-01-01 00:15 then the first time point after aggregation will be 2011-01-01 01:00. For the ELECTRICITY dataset we removed the first year from training set, to match the training set used in (Yu et al., 2016), based on the aggregated dataset downloaded from, presumable authors', github repository[4]. We also made sure that data points for both ELECTRICITY and TRAFFIC datasets after aggregation match those used in (Yu et al., 2016). The authors of MatFact model were using the last 7 days of datasets as test set, but papers from Amazon are using different splits, where the split points are provided by a date. Changing split points without a well grounded reason adds uncertainties to the comparability of the models performances and creates challenges to the reproducibility of the results, thus we were trying to match all different splits in our experiments. It was especially challenging on TRAFFIC dataset, where we had to use some heuristics to find records dates; the dataset authors state: " The measurements cover the period from Jan. 1st 2008 to Mar. 30th 2009" and " We remove public holidays from the dataset, as well as two days with anomalies (March 8th 2009 and March 9th 2008) where all sensors were muted between 2:00 and 3:00 AM. " , but we failed to match a part of the provided labels of week days to actual dates. Therefore, we had to assume that the actual list of gaps, which include holidays and anomalous days, is the following:

1. Jan. 1, 2008 (New Year's Day)
2. Jan. 21, 2008 (Martin Luther King Jr. Day)
3. Feb. 18, 2008 (Washington's Birthday)
4. Mar. 9, 2008 (Anomaly day)
5. May 26, 2008 (Memorial Day)
6. Jul. 4, 2008 (Independence Day)
7. Sep. 1, 2008 (Labor Day)
8. Oct. 13, 2008 (Columbus Day)
9. Nov. 11, 2008 (Veterans Day)
10. Nov. 27, 2008 (Thanksgiving)
11. Dec. 25, 2008 (Christmas Day)
12. Jan. 1, 2009 (New Year's Day)
13. Jan. 19, 2009 (Martin Luther King Jr. Day)
14. Feb. 16, 2009 (Washington's Birthday)
15. Mar. 8, 2009 (Anomaly day)

The first 6 gaps were confirmed by the gaps in labels, but the rest were more than 1 day apart from any public holiday of years 2008 and 2009 in San Francisco, California and US. More over the number of gaps we found in the labels provided by dataset authors is 10, while the number of days between Jan. 1st 2008 and Mar. 30th 2009 is 455, assuming that Jan. 1st 2008 was skipped from the values and labels we should end up with either $454 - 10 = 444$ instead of 440 days or different end date.

The metric is reported in Normalized deviation (ND) as in (Yu et al., 2016) which is equal to $p50$ loss used in DeepAR, Deep State, and Deep Factors papers.

---

[2]https://archive.ics.uci.edu/ml/datasets/ElectricityLoadDiagrams20112014
[3]https://archive.ics.uci.edu/ml/datasets/PEMS-SF
[4]https://github.com/rofuyu/exp-trmf-nips16/blob/master/python/exp-scripts/datasets/download-data.sh

$$ND = \frac{\sum_{i,t} |\hat{Y}_{it} - Y_{it}|}{\sum_{i,t} |Y_{it}|} \tag{7}$$

Table 16: ND Performance on the ELECTRICITY and TRAFFIC test sets.
[1] Split used in DeepAR (Flunkert et al., 2017) and Deep State (Rangapuram et al., 2018a).
[2] Split used in Deep Factors (Wang et al., 2019).
[†] Numbers reported by (Flunkert et al., 2017), which are different from the original MatFact paper, hypothetically due to changed split point.

| | ELECTRICITY | | | TRAFFIC | | |
| | 2014-09-01[1] | 2014-03-31[2] | last 7 days | 2008-06-15[1] | 2008-01-14[2] | last 7 days |
|---|---|---|---|---|---|---|
| MatFact | 0.16[†] | n/a | 0.255 | 0.20[†] | n/a | 0.187 |
| DeepAR | 0.07 | 0.272 | n/a | 0.17 | 0.296 | n/a |
| Deep State | 0.083 | n/a | n/a | 0.167 | n/a | n/a |
| Deep Factors | n/a | 0.112 | n/a | n/a | **0.225** | n/a |
| N-BEATS-G (ours) | **0.064** | **0.065** | **0.171** | **0.114** | 0.230 | 0.112 |
| N-BEATS-I (ours) | 0.073 | 0.072 | 0.185 | **0.114** | 0.231 | **0.110** |
| N-BEATS-I+G (ours) | 0.067 | 0.067 | 0.178 | **0.114** | 0.230 | 0.111 |

Contrary to Amazon models N-BEATS does not use any covariates, like day-of-week, hour-of-day, etc.

The N-BEATS architecture used in this experiment is exactly the same as used in M4, M3 and TOURISM datasets, the only difference is history size and the number of iterations. These parameters were chosen based on performance on validation set. Where the validation set consists of 7 consecutive days right before the test set. After the parameters are chosen the model is retrained on training set which includes the validation set, then tested on test set. The model is trained once and tested on test set using rolling window operation described in (Yu et al., 2016).

## C.5 DETAILED RESULTS: COMPARE TO DEEPAR, DEEP STATE SPACE MODELS

Table 17 compares ND (7) performance of DeepAR, DeepState models published in (Rangapuram et al., 2018a) and N-BEATS.

Table 17: ND Performance of DeepAR, Deep State Space, and N-BEATS models on M4-Hourly and TOURISM datasets

|                    | M4 (Hourly) | TOURISM (Monthly) | TOURISM (Quarterly) |
|--------------------|-------------|-------------------|---------------------|
| DeepAR             | 0.09        | 0.107             | 0.11                |
| DeepState          | 0.044       | 0.138             | 0.098               |
| N-BEATS-G (ours)   | **0.023**   | **0.097**         | 0.080               |
| N-BEATS-I (ours)   | 0.027       | 0.103             | 0.079               |
| N-BEATS-I+G (ours) | 0.025       | 0.099             | **0.077**           |

Table 18: Settings of hyperparameters across subsets of M4, M3, TOURISM datasets.

| | M4 | | | | | | M3 | | | | TOURISM | | |
|---|---|---|---|---|---|---|---|---|---|---|---|---|---|
| | Yly | Qly | Mly | Wly | Dly | Hly | Yly | Qly | Mly | Other | Yly | Qly | Mly |
| **Parameter** | | | | | | | N-BEATS-I | | | | | | |
| $L_H$ | 1.5 | 1.5 | 1.5 | 10 | 10 | 10 | 20 | 5 | 5 | 20 | 20 | 10 | 20 |
| Iterations | 15K | 15K | 15K | 5K | 5K | 5K | 50 | 6K | 6K | 250 | 30 | 500 | 300 |
| Losses | SMAPE/MAPE/MASE | | | | | | SMAPE/MAPE/MASE | | | | MAPE | | |
| S-width | 2048 | | | | | | | | | | | | |
| S-blocks | 3 | | | | | | | | | | | | |
| S-block-layers | 4 | | | | | | | | | | | | |
| T-width | 256 | | | | | | | | | | | | |
| T-degree | 2 | | | | | | | | | | | | |
| T-blocks | 3 | | | | | | | | | | | | |
| T-block-layers | 4 | | | | | | | | | | | | |
| Sharing | STACK LEVEL | | | | | | | | | | | | |
| Lookback period | $2H,3H,4H,5H,6H,7H$ | | | | | | | | | | | | |
| Batch | 1024 | | | | | | | | | | | | |
| **Parameter** | | | | | | | N-BEATS-G | | | | | | |
| $L_H$ | 1.5 | 1.5 | 1.5 | 10 | 10 | 10 | 20 | 20 | 20 | 10 | 5 | 10 | 20 |
| Iterations | 15K | 15K | 15K | 5K | 5K | 5K | 20 | 250 | 10K | 250 | 30 | 100 | 100 |
| Losses | SMAPE/MAPE/MASE | | | | | | SMAPE/MAPE/MASE | | | | MAPE | | |
| Width | 512 | | | | | | | | | | | | |
| Blocks | 1 | | | | | | | | | | | | |
| Block-layers | 4 | | | | | | | | | | | | |
| Stacks | 30 | | | | | | | | | | | | |
| Sharing | NO | | | | | | | | | | | | |
| Lookback period | $2H,3H,4H,5H,6H,7H$ | | | | | | | | | | | | |
| Batch | 1024 | | | | | | | | | | | | |

## D   HYPER-PARAMETER SETTINGS

Table 18 presents the hyperparameter settings used to train models on different subsets of M4, M3 and TOURISM datasets. A brief discussion of field names in the table is warranted.

Subset names **Yly, Qly, Mly, Wly, Dly, Hly, Other** correspond to yearly, quarterly, monthly, weekly, daily, hourly and other frequency subsets defined in the original datasets.

**N-BEATS-I** and **N-BEATS-G** correspond to the interpretable and generic model configurations defined in Section 3.3.

### D.1   COMMON PARAMETERS

$L_H$ is the coefficient defining the length of training history immediately preceding the last point in the train part of the TS that is used to generate training samples. For example, if for M4 Yearly the forecast horizon is 6 and $L_H$ is 1.5, then we consider $1.5 \cdot 6 = 9$ most recent points in the train dataset for each time series to generate training samples. A training sample from a given TS in M4 Yearly is then generated by choosing one of the most recent 9 points as an anchor. All the points preceding the anchor are used to create the input to N-BEATS, while the points following and including the anchor become training target. Target and history points that fall outside of the time series limits given the anchor position are filled with zeros and masked during the training. We observed that for subsets with large number of time series $L_H$ tends to be smaller and for subsets with smaller number of time series it tends to be larger. For example, in massive Yearly, Monthly, Quarterly subsets of M4 $L_H$ is equal to 1.5; and in moderate to small Weekly, Daily, Hourly subsets of M4 $L_H$ is equal to 10.

**Iterations** is the number of batches used to train N-BEATS.

**Losses** is the set of loss functions that is used to build ensemble. We observed on the respective validation sets that for M4 and M3 mixing models trained on a variety of metrics resulted in performance gain. In the case of TOURISM dataset training only on MAPE led to the best validation scores.

**Sharing** defines whether the coefficients in the fully-connected layers are shared. We observed that the interpretable model works best when weights are shared across stack, while generic model works best when none of the weights are shared.

**Lookback period** is the length of the history window forming the input to the model (please refer to Figure 1). This is the function of the forecast horizon length, $H$. In our experiments we mixed models with lookback periods $2H, 3H, 4H, 5H, 6H, 7H$ in one ensemble. As an example, for a forecast horizon length $H = 8$ and a lookback period $7H$, the model's input will consist of the history window of $7 \cdot 8 = 56$ samples.

**Batch** is the batch size. We used batch size of 1024. We observed that the training was faster with larger batch sizes, however in our setup little gain was observed with batch sizes beyond 1024.

### D.2   N-BEATS-I PARAMETERS

**S-width** is the width of the fully connected layers in the blocks comprising the seasonality stack of the interpretable model (please refer to Figure 1).

**S-blocks** is the number of blocks comprising the seasonality stack of the interpretable model (please refer to Figure 1).

**S-block-layers** is the number of fully-connected layers comprising one block in the seasonality stack of the interpretable model (preceding the final fully-connected projection layers forming the backcast/forecast fork, please refer to Figure 1).

**T-width** is the width of the fully connected layers in the blocks comprising the trend stack of the interpretable model (please refer to Figure 1).

**T-degree** is the degree $p$ of polynomial in the trend stack of the interpretable model (please refer to equation (2)).

**T-blocks** is the number of blocks comprising the trend stack of the interpretable model (please refer to Figure 1).

**T-block-layers** is the number of fully-connected layers comprising one block in the trend stack of the interpretable model (preceding the final fully-connected projection layers forming the backcast/forecast fork, please refer to Figure 1).

### D.3   N-BEATS-G PARAMETERS

**Width** is the width of the fully connected layers in the blocks comprising the stacks of the generic model (please refer to Figure 1).

**Blocks** is the number of blocks comprising the stack of the generic model (please refer to Figure 1).

**Block-layers** is the number of fully-connected layers comprising one block in the stack of the generic model (preceding the final fully-connected projection layers forming the backcast/forecast fork, please refer to Figure 1).

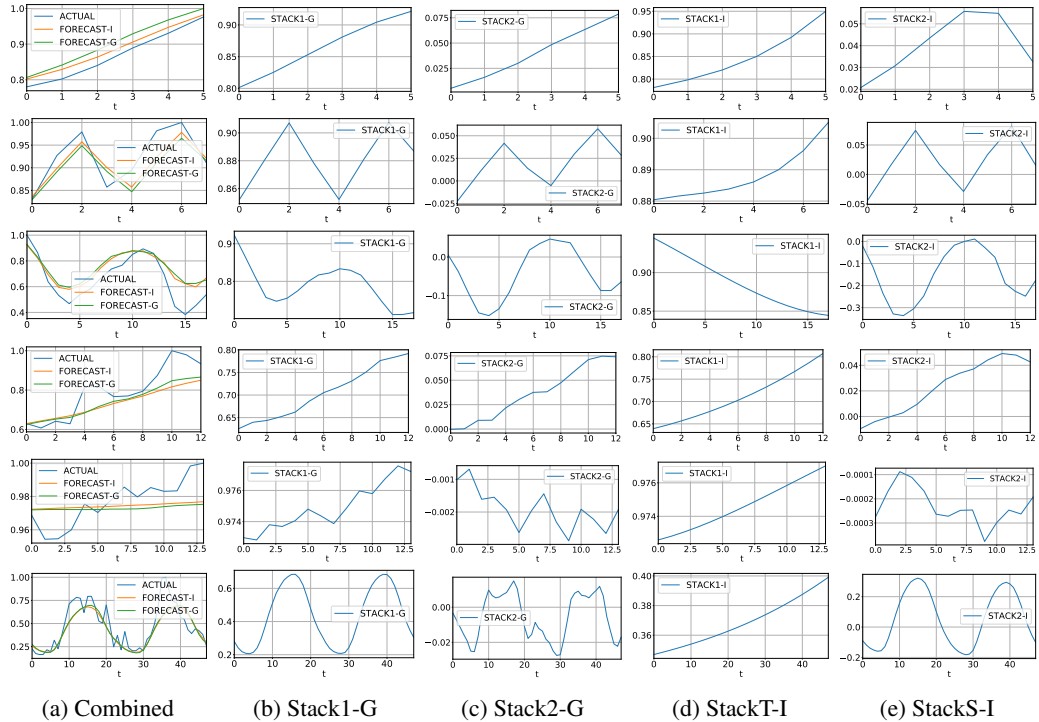

| (a) Combined | (b) Stack1-G | (c) Stack2-G | (d) StackT-I | (e) StackS-I |

Figure 5: The outputs of generic and the interpretable configurations, M4 dataset. Each row is one time series example per data frequency, top to bottom (Yearly: id Y3974, Quarterly: id Q11588, Monthly: id M19006, Weekly: id W246, Daily: id D404, Hourly: id H344). The magnitudes in a row are normalized by the maximal value of the actual time series for convenience. Column (a) shows the actual values (ACTUAL), the generic model forecast (FORECAST-G) and the interpretable model forecast (FORECAST-I). Columns (b) and (c) show the outputs of stacks 1 and 2 of the generic model, respectively; FORECAST-G is their summation. Columns (d) and (e) show the output of the Trend and the Seasonality stacks of the interpretable model, respectively; FORECAST-I is their summation.

# E  DETAILED SIGNAL TRACES OF INTERPRETABLE INPUTS PRESENTED IN FIGURE 2

The goal of this section is to show the detailed traces (numeric values) of signals visualized in Fig. 2. This is to demonstrate that even though the StackT-I (Fig. 2 (d)) and StackS-I (Fig. 2 (e)) provide response lines different from the counterparts in Stack1-G (Fig. 2 (b)) and Stack2-G (Fig. 2 (c)), the summations in the combined line (Fig. 2 (a)) can still be very similar.

First, we reproduce Fig. 5 for the convenience of the reader. Second, for each row in the figure, we produce a table showing the numeric values of each signal depicted in corresponding plots (please refer to Tables 19– 24). We make sure that the names of signals in figure legends and in the table columns match, such that they can easily be cross-referenced. It can be clearly seen in Tables 19– 24 that (i) traces STACK1-I and STACK2-I sum up to trace FORECAST-I, (ii) traces STACK1-G and STACK2-G sum up to trace FORECAST-G, (iii) traces FORECAST-I and FORECAST-G are overall very similar even though their components may significantly differ from each other.

Table 19: Detailed traces of signals depicted in row 1 of Fig. 5, corresponding to the time series Yearly: id Y3974.

| t | ACTUAL | FORECAST-I | FORECAST-G | STACK1-I | STACK2-I | STACK1-G | STACK2-G |
|---|--------|-----------|-----------|----------|----------|----------|----------|
| 0 | 0.780182 | 0.802068 | 0.806608 | 0.781290 | 0.020778 | 0.801294 | 0.005314 |
| 1 | 0.802337 | 0.829223 | 0.841406 | 0.798422 | 0.030801 | 0.825271 | 0.016135 |
| 2 | 0.840317 | 0.863683 | 0.883136 | 0.820196 | 0.043487 | 0.853114 | 0.030022 |
| 3 | 0.889376 | 0.905962 | 0.929258 | 0.850250 | 0.055712 | 0.880833 | 0.048425 |
| 4 | 0.930521 | 0.947028 | 0.967846 | 0.892221 | 0.054807 | 0.904393 | 0.063453 |
| 5 | 0.976414 | 0.982307 | 1.000000 | 0.949748 | 0.032559 | 0.921360 | 0.078640 |

Table 20: Detailed traces of signals depicted in row 2 of Fig. 5, corresponding to the time series Quarterly: id Q11588.

| t | ACTUAL | FORECAST-I | FORECAST-G | STACK1-I | STACK2-I | STACK1-G | STACK2-G |
|---|--------|-----------|-----------|----------|----------|----------|----------|
| 0 | 0.830068 | 0.835964 | 0.829417 | 0.880435 | -0.044471 | 0.852018 | -0.022601 |
| 1 | 0.927155 | 0.898949 | 0.891168 | 0.881626 | 0.017324 | 0.880124 | 0.011044 |
| 2 | 0.979204 | 0.957379 | 0.948799 | 0.882549 | 0.074831 | 0.907149 | 0.041650 |
| 3 | 0.857250 | 0.900612 | 0.891967 | 0.883830 | 0.016782 | 0.877959 | 0.014008 |
| 4 | 0.895082 | 0.857230 | 0.847029 | 0.886096 | -0.028866 | 0.852232 | -0.005204 |
| 5 | 0.981590 | 0.923832 | 0.911001 | 0.889972 | 0.033860 | 0.881140 | 0.029861 |
| 6 | 1.000000 | 0.978128 | 0.965236 | 0.896085 | 0.082043 | 0.907475 | 0.057761 |
| 7 | 0.910528 | 0.920632 | 0.915460 | 0.905062 | 0.015571 | 0.886941 | 0.028519 |

Table 21: Detailed traces of signals depicted in row 3 of Fig. 5, corresponding to the time series Monthly: id M19006.

| t | ACTUAL | FORECAST-I | FORECAST-G | STACK1-I | STACK2-I | STACK1-G | STACK2-G |
|---|--------|-----------|-----------|----------|----------|----------|----------|
| 0 | 1.000000 | 0.923394 | 0.928279 | 0.944660 | -0.021266 | 0.922835 | 0.005444 |
| 1 | 0.865248 | 0.822588 | 0.829924 | 0.937575 | -0.114987 | 0.867619 | -0.037695 |
| 2 | 0.638298 | 0.693820 | 0.717119 | 0.930295 | -0.236475 | 0.810818 | -0.093699 |
| 3 | 0.531915 | 0.594375 | 0.612377 | 0.922890 | -0.328515 | 0.757199 | -0.144823 |
| 4 | 0.468085 | 0.579403 | 0.595221 | 0.915428 | -0.336025 | 0.747151 | -0.151930 |
| 5 | 0.539007 | 0.602615 | 0.620809 | 0.907977 | -0.305362 | 0.755078 | -0.134269 |
| 6 | 0.581560 | 0.653387 | 0.682669 | 0.900606 | -0.247219 | 0.774561 | -0.091891 |
| 7 | 0.666667 | 0.747440 | 0.765814 | 0.893385 | -0.145945 | 0.799594 | -0.033781 |
| 8 | 0.737589 | 0.817883 | 0.835577 | 0.886382 | -0.068498 | 0.817218 | 0.018359 |
| 9 | 0.765957 | 0.862568 | 0.856962 | 0.879665 | -0.017097 | 0.822099 | 0.034862 |
| 10 | 0.851064 | 0.873448 | 0.880074 | 0.873304 | 0.000145 | 0.833473 | 0.046601 |
| 11 | 0.893617 | 0.878186 | 0.871103 | 0.867367 | 0.010819 | 0.829537 | 0.041566 |
| 12 | 0.858156 | 0.834448 | 0.853549 | 0.861923 | -0.027475 | 0.816527 | 0.037022 |
| 13 | 0.695035 | 0.785341 | 0.776687 | 0.857040 | -0.071699 | 0.782536 | -0.005850 |
| 14 | 0.446809 | 0.662443 | 0.697788 | 0.852789 | -0.190345 | 0.745623 | -0.047835 |
| 15 | 0.382979 | 0.623196 | 0.624614 | 0.849236 | -0.226040 | 0.711553 | -0.086939 |
| 16 | 0.453901 | 0.598511 | 0.625150 | 0.846451 | -0.247941 | 0.712130 | -0.086980 |
| 17 | 0.539007 | 0.668231 | 0.652175 | 0.844504 | -0.176272 | 0.716925 | -0.064750 |

Table 22: Detailed traces of signals depicted in row 4 of Fig. 5, corresponding to the time series Weekly: id W246.

| t | ACTUAL | FORECAST-I | FORECAST-G | STACK1-I | STACK2-I | STACK1-G | STACK2-G |
|---|--------|-----------|-----------|----------|----------|----------|----------|
| 0 | 0.630056 | 0.629703 | 0.625108 | 0.639236 | -0.009534 | 0.625416 | -0.000309 |
| 1 | 0.607536 | 0.643509 | 0.639846 | 0.647549 | -0.004039 | 0.639592 | 0.000254 |
| 2 | 0.641731 | 0.656171 | 0.652584 | 0.656696 | -0.000526 | 0.643665 | 0.008919 |
| 3 | 0.628783 | 0.669636 | 0.661163 | 0.666739 | 0.002897 | 0.652107 | 0.009056 |
| 4 | 0.816799 | 0.687287 | 0.683860 | 0.677738 | 0.009549 | 0.662176 | 0.021683 |
| 5 | 0.817020 | 0.709211 | 0.717187 | 0.689752 | 0.019459 | 0.686589 | 0.030598 |
| 6 | 0.766724 | 0.731732 | 0.742824 | 0.702841 | 0.028891 | 0.705234 | 0.037590 |
| 7 | 0.770320 | 0.750834 | 0.755154 | 0.717066 | 0.033768 | 0.716986 | 0.038167 |
| 8 | 0.794113 | 0.769671 | 0.778460 | 0.732487 | 0.037184 | 0.731113 | 0.047347 |
| 9 | 0.874011 | 0.793373 | 0.810332 | 0.749164 | 0.044209 | 0.750939 | 0.059392 |
| 10 | 1.000000 | 0.816386 | 0.847545 | 0.767157 | 0.049229 | 0.776405 | 0.071140 |
| 11 | 0.979251 | 0.834532 | 0.858604 | 0.786526 | 0.048006 | 0.783939 | 0.074665 |
| 12 | 0.933160 | 0.850010 | 0.866116 | 0.807332 | 0.042678 | 0.792134 | 0.073982 |

Table 23: Detailed traces of signals depicted in row 5 of Fig. 5, corresponding to the time series Daily: id D404.

| t | ACTUAL | FORECAST-I | FORECAST-G | STACK1-I | STACK2-I | STACK1-G | STACK2-G |
|---|--------|-----------|-----------|----------|----------|----------|----------|
| 0 | 0.968704 | 0.972314 | 0.971950 | 0.972589 | -0.000275 | 0.972964 | -0.001014 |
| 1 | 0.954319 | 0.972637 | 0.972131 | 0.972808 | -0.000171 | 0.972822 | -0.000690 |
| 2 | 0.954599 | 0.972972 | 0.972188 | 0.973060 | -0.000088 | 0.973798 | -0.001610 |
| 3 | 0.959959 | 0.973230 | 0.972140 | 0.973341 | -0.000112 | 0.973686 | -0.001546 |
| 4 | 0.975472 | 0.973481 | 0.972125 | 0.973649 | -0.000168 | 0.974060 | -0.001934 |
| 5 | 0.970391 | 0.973715 | 0.972174 | 0.973979 | -0.000264 | 0.974800 | -0.002626 |
| 6 | 0.977728 | 0.974056 | 0.972403 | 0.974328 | -0.000272 | 0.974368 | -0.001965 |
| 7 | 0.985624 | 0.974445 | 0.972428 | 0.974693 | -0.000248 | 0.973870 | -0.001442 |
| 8 | 0.979695 | 0.974823 | 0.972567 | 0.975069 | -0.000246 | 0.974870 | -0.002303 |
| 9 | 0.985345 | 0.975079 | 0.973089 | 0.975455 | -0.000376 | 0.975970 | -0.002881 |
| 10 | 0.983088 | 0.975547 | 0.973881 | 0.975845 | -0.000298 | 0.975796 | -0.001915 |
| 11 | 0.983368 | 0.975991 | 0.974537 | 0.976238 | -0.000247 | 0.976757 | -0.002220 |
| 12 | 0.998312 | 0.976365 | 0.974924 | 0.976628 | -0.000263 | 0.977579 | -0.002655 |
| 13 | 1.000000 | 0.976821 | 0.975291 | 0.977013 | -0.000193 | 0.977213 | -0.001922 |

Table 24: Detailed traces of signals depicted in row 6 of Fig. 5, corresponding to the time series Hourly: id H344.

| t | ACTUAL | FORECAST-I | FORECAST-G | STACK1-I | STACK2-I | STACK1-G | STACK2-G |
|---|--------|-----------|-----------|----------|----------|----------|----------|
| 0 | 0.226804 | 0.256799 | 0.277159 | 0.346977 | -0.090179 | 0.280489 | -0.003329 |
| 1 | 0.175258 | 0.228913 | 0.234605 | 0.347615 | -0.118701 | 0.241790 | -0.007185 |
| 2 | 0.164948 | 0.209208 | 0.207347 | 0.348265 | -0.139057 | 0.218575 | -0.011228 |
| 3 | 0.164948 | 0.197360 | 0.193084 | 0.348928 | -0.151568 | 0.208458 | -0.015374 |
| 4 | 0.216495 | 0.190397 | 0.186586 | 0.349606 | -0.159209 | 0.205701 | -0.019115 |
| 5 | 0.195876 | 0.194204 | 0.189433 | 0.350297 | -0.156094 | 0.214399 | -0.024966 |
| 6 | 0.319588 | 0.221026 | 0.216221 | 0.351004 | -0.129978 | 0.241574 | -0.025353 |
| 7 | 0.226804 | 0.279857 | 0.276414 | 0.351726 | -0.071869 | 0.293580 | -0.017167 |
| 8 | 0.371134 | 0.357292 | 0.359372 | 0.352464 | 0.004828 | 0.364392 | -0.005020 |
| 9 | 0.536082 | 0.438540 | 0.446126 | 0.353218 | 0.085322 | 0.442703 | 0.003423 |
| 10 | 0.711340 | 0.511441 | 0.519928 | 0.353989 | 0.157452 | 0.510142 | 0.009787 |
| 11 | 0.752577 | 0.571604 | 0.578186 | 0.354777 | 0.216827 | 0.571596 | 0.006590 |
| 12 | 0.783505 | 0.617085 | 0.618778 | 0.355584 | 0.261501 | 0.613425 | 0.005353 |
| 13 | 0.773196 | 0.651777 | 0.655123 | 0.356409 | 0.295368 | 0.649259 | 0.005864 |
| 14 | 0.618557 | 0.670202 | 0.676814 | 0.357253 | 0.312950 | 0.669555 | 0.007260 |
| 15 | 0.793814 | 0.679884 | 0.692592 | 0.358116 | 0.321768 | 0.684208 | 0.008384 |
| 16 | 0.793814 | 0.672488 | 0.696440 | 0.359000 | 0.313488 | 0.684764 | 0.011676 |
| 17 | 0.680412 | 0.648851 | 0.677696 | 0.359904 | 0.288947 | 0.662714 | 0.014983 |
| 18 | 0.525773 | 0.602496 | 0.630922 | 0.360828 | 0.241667 | 0.620368 | 0.010554 |
| 19 | 0.505155 | 0.537698 | 0.552296 | 0.361775 | 0.175923 | 0.552599 | -0.000304 |
| 20 | 0.701031 | 0.463760 | 0.466442 | 0.362743 | 0.101016 | 0.477429 | -0.010987 |
| 21 | 0.484536 | 0.395795 | 0.390958 | 0.363734 | 0.032061 | 0.408708 | -0.017750 |
| 22 | 0.247423 | 0.337809 | 0.338500 | 0.364748 | -0.026939 | 0.354028 | -0.015528 |
| 23 | 0.371134 | 0.292452 | 0.303902 | 0.365786 | -0.073334 | 0.312588 | -0.008686 |
| 24 | 0.216495 | 0.254359 | 0.258435 | 0.366848 | -0.112489 | 0.270568 | -0.012133 |
| 25 | 0.412371 | 0.227557 | 0.224291 | 0.367934 | -0.140377 | 0.237846 | -0.013555 |
| 26 | 0.237113 | 0.207962 | 0.201250 | 0.369046 | -0.161084 | 0.219420 | -0.018169 |
| 27 | 0.206186 | 0.196049 | 0.189439 | 0.370183 | -0.174133 | 0.209743 | -0.020304 |
| 28 | 0.206186 | 0.189030 | 0.182843 | 0.371346 | -0.182316 | 0.207727 | -0.024884 |
| 29 | 0.237113 | 0.194524 | 0.185734 | 0.372536 | -0.178011 | 0.213194 | -0.027460 |
| 30 | 0.206186 | 0.220227 | 0.215444 | 0.373753 | -0.153526 | 0.242485 | -0.027041 |
| 31 | 0.329897 | 0.279614 | 0.274624 | 0.374998 | -0.095383 | 0.292834 | -0.018210 |
| 32 | 0.371134 | 0.355078 | 0.358020 | 0.376270 | -0.021193 | 0.365332 | -0.007312 |
| 33 | 0.494845 | 0.437103 | 0.445832 | 0.377572 | 0.059531 | 0.441323 | 0.004510 |
| 34 | 0.690722 | 0.509515 | 0.520006 | 0.378903 | 0.130612 | 0.512064 | 0.007942 |
| 35 | 0.989691 | 0.570761 | 0.579003 | 0.380263 | 0.190497 | 0.569851 | 0.009152 |
| 36 | 1.000000 | 0.615868 | 0.623981 | 0.381654 | 0.234214 | 0.617254 | 0.006728 |
| 37 | 0.845361 | 0.651487 | 0.656782 | 0.383076 | 0.268411 | 0.650336 | 0.006446 |
| 38 | 0.742268 | 0.670664 | 0.678412 | 0.384528 | 0.286136 | 0.673055 | 0.005357 |
| 39 | 0.721649 | 0.680534 | 0.691961 | 0.386013 | 0.294521 | 0.684347 | 0.007614 |
| 40 | 0.567010 | 0.671607 | 0.692853 | 0.387530 | 0.284078 | 0.683297 | 0.009555 |
| 41 | 0.546392 | 0.648851 | 0.672476 | 0.389079 | 0.259771 | 0.660613 | 0.011863 |
| 42 | 0.432990 | 0.599785 | 0.621940 | 0.390662 | 0.209123 | 0.615426 | 0.006514 |
| 43 | 0.391753 | 0.537520 | 0.544543 | 0.392279 | 0.145241 | 0.549961 | -0.005417 |
| 44 | 0.443299 | 0.462772 | 0.457700 | 0.393930 | 0.068842 | 0.471080 | -0.013380 |
| 45 | 0.422680 | 0.397098 | 0.380324 | 0.395616 | 0.001482 | 0.401229 | -0.020905 |
| 46 | 0.381443 | 0.342213 | 0.325583 | 0.397337 | -0.055124 | 0.347827 | -0.022244 |
| 47 | 0.257732 | 0.297711 | 0.287130 | 0.399094 | -0.101384 | 0.304270 | -0.017140 |

