# OpenReview forum: "N-BEATS: Neural basis expansion analysis for interpretable time series forecasting"
_ICLR.cc/2020/Conference — Accept (Poster)_

### Official Review · AnonReviewer1 · 2019-10-24
**Official Blind Review #1**

**Rating:** 8

**Review:**

This goal of this paper is to present a strong empirical result showing that a "pure" machine learning based method can outperform all known methods on some of the most challenging time series forecasting benchmarks (TOURISM, M3 and especially M4). Since I am not from the field of forecasting, I can not be sure of this, but from my understanding these benchmark datasets are indeed challenging and the cited references back up the claims of the paper related to these datasets being important in the field.

On the most challenging dataset (M4), the best known performing method combines RNNs with a traditional smoothing algorithm. The model proposed in this paper outperforms it without being combined with any classical approach, though it does utilize ensembling.
The experimental setup is sound in my opinion, and the result appears to be of high potential significance.
However, despite trying to go through Section 3 multiple times, the exact model architecture is not clear to me. Due to this reason, my current decision is a weak rejection since the model is a central contribution of the paper. I will be happy to increase my score if the authors can make the model description crystal clear.

Even though my expertise is deep neural network architectures, I find it hard to follow the descriptions in Sec. 3. I faced the most difficulty understanding section 3.1, which obviously made the rest of subsections even harder to follow. Here are my main points of confusion:

- One big issue is that the paper uses an illustration (Fig. 1) to explain the architecture instead of equations, but then uses symbols in the main text that do not appear on Fig. 1 at all such as g_theta. Is the "FC Stack (4 layers)" g_theta?

- Where are (uppercase) phi functions? I could infer that these are the "FC" blocks but they should be labeled.

- The Figure has the symbols g^b_theta and g^f_theta that do not appear in the text description. What exactly do they do? And is the theta that parameterizes each of them the same theta that parameterizes g_theta? How is this possible if g_theta is the "FC Stack (4 layers)"?

- The description in second and third paragraph of Section 3.1 is very confusing and unclear. It should be replaced or augmented with equations using clearly defined symbols that match Figure 1.

- More confusion stems from the use of the term "parameters" in (I believe) a different context than is used in neural networks, where "parameters" refers to connection weights. But here parameters are outputs of some functions, so either they are not connection weights or this is a fast-weight style architecture where outputs are weights [1], in which this should be made clear.

- Design of the doubly residual architecture in Section 3.2 makes sense to me at a high level, but I feel it is still very hard to clearly understand and implement it. Again, use of equations to clearly define the computation would be very helpful.


[1] Schmidhuber, Jürgen. "Learning to control fast-weight memories: An alternative to dynamic recurrent networks." Neural Computation 4.1 (1992): 131-139.


--- Update after rebuttal ---

I am happy to see the paper greatly improved by the authors in their updates. My concerns related to the presentation of the model have been addressed, and I find the architecture much easier to understand. I also appreciate the detailed supplementary material that is likely to help readers interested in the area. Related to the areas I work in, I noticed the following missing references:

Densenets: Lang, K.J. and Witbrock, M.J., 1988, June. Learning to tell two spirals apart. In Proceedings of the 1988 connectionist models summer school (No. 1989, pp. 52-59).

Metalearning: Schmidhuber, J., 1987. Evolutionary principles in self-referential learning, or on learning how to learn: the meta-meta-... hook (Doctoral dissertation, Technische Universität München).

While improving papers is generally the objective of the rebuttal phase, I suggest that the authors to not take this as an opportunity to submit unpolished papers in the first phase. That said, I have increased my rating to reflect my satisfaction with the current version of the paper.


**Experience Assessment:**

I do not know much about this area.

**Review Assessment: Checking Correctness Of Derivations And Theory:**

N/A

**Review Assessment: Checking Correctness Of Experiments:**

I assessed the sensibility of the experiments.

**Review Assessment: Thoroughness In Paper Reading:**

I read the paper thoroughly.

---

> ### Author Response · Authors · 2019-11-05
> **Equations will be added**
>
> We would like to sincerely thank Reviewer 1 for the thorough analysis of the manuscript and for insightful feedback. We will add equations to describe the operation of the architecture in more detail, as suggested by the Reviewer. In the meanwhile, we would like to ask Reviewer's advice on the placement of the equations. Would it be better to have a separate section with mathematical equations in the appendix, or embedding the equations directly in the text of Section 3 is a better choice?

---

> > ### Comment · AnonReviewer1 · 2019-11-09
> > **Suggestion**
> >
> > Dear authors,
> >
> > I think equations and corresponding diagrams to explain the model as clearly as possible should be presented together.  Of course, since you are working with limited space, you may need to shorten some other sections or the diagrams, and then add larger versions of those to the appendices.

---

> > > ### Author Response · Authors · 2019-11-10
> > > **Response to Reviewer 1**
> > >
> > > We would like to sincerely thank Reviewer 1 once again for providing additional feedback on the placement of equations as well as for the insightful comments posted in the review. We have included the additional equations directly in the revised section 3, as advised by the Reviewer. We believe that we fully addressed the concerns raised by Reviewer 1 and this significantly improved the clarity of the paper in general, and Section 3 specifically. At the same time, we would be more than happy to further adjust the paper should Reviewer 1 have any additional comments, questions or concerns. Our more detailed response is provided below.
> > >
> > > 1. We made sure that the notation in Figure 1 is aligned with the rest of the paper. Any differences between the figure and the text are explicitly pointed out and explained in the text.
> > >
> > > 2. The exact nature of basis functions g is explained via equations to make it clear.
> > >
> > > 3. The entire architecture is explained in detail via mathematical equations, both at the block level and at the higher level of doubly residual topology.
> > >
> > > 4. We made clear that \theta are the expansion coefficients. As Reviewer 1 mentioned, \theta are not connection weights, and calling them parameters may be confusing.  Therefore, instead of referring to \theta as parameters, we refer to them as expansion coefficients throughout the paper now.

---

### Official Review · AnonReviewer3 · 2019-10-25
**Official Blind Review #3**

**Rating:** 6

**Review:**

The paper investigates a pure deep learning architecture for Univariate time series analysis by simply ensembling feed-forward networks, along with the residual stacking mechanism for fluid learning. Each of the generic block consists of 4 FC layers followed by the use of forward and backward predictor to have forecast and backcast output of the original input. These blocks forms the stacks, where each stack provides the residuals and the forecast responses further to the next stacks, which ultimately provide the global forecast. To make the internal stack outputs interpretable, assumptions are imposed on the trend model, which follows a polynomial function of time vector, and seasonality model which follows periodic Fourier series. Further, the ensembling of models based on different metrics and input windows is used for better accuracy.

Very well written paper and easy to follow. It advocates a pure DL framework (instead of hybrid statistical models and DL). I found the idea simple and effective, yielding results better than the previous approaches. Also, the experimental setup is well described. I also found the link between this work and meta-learning approaches interesting.

However, I have several questions about the interpretability results. It looks like the inductive bias based on some general assumptions can fail in some cases. For example, in Fig. 2(a), the line for FORECAST-I and FORECAST-G deviates much in case of hourly/weekly/daily data frequency. What is the reason behind this?

Also, in Fig. 2, while the StackT-I (fig.2(d)) and StackS-I (fig.2(e)) provide response lines different from the counterparts in  Stack1-G (fig.2(b)) and Stack2-G (fig.2(c)), the summations in the combined line (fig.2(a)) yield similar curves of pretty much the same shapes, without much perceived difference. Is it expected or something is wrong?


**Experience Assessment:**

I do not know much about this area.

**Review Assessment: Checking Correctness Of Derivations And Theory:**

I assessed the sensibility of the derivations and theory.

**Review Assessment: Checking Correctness Of Experiments:**

I assessed the sensibility of the experiments.

**Review Assessment: Thoroughness In Paper Reading:**

I read the paper at least twice and used my best judgement in assessing the paper.

---

> ### Author Response · Authors · 2019-11-05
> **More discussion of Figure 2 will be added**
>
> We would like to sincerely thank Reviewer 3 for the thorough analysis of the manuscript and for insightful feedback. We will extend the discussion of Figure 2 to thoroughly address the concerns raised by the reviewer.
>
> In the meantime, we would like to clarify if  we should interpret "the line for FORECAST-I and FORECAST-G deviates much" that the forecast provided by FORECAST-I is different from the forecast provided by FORECAST-G?

---

> ### Author Response · Authors · 2019-11-12
> **Response to Reviewer 3**
>
> First of all, we would like to thank Reviewer 3 for the thorough review of the manuscript and for insightful questions. We believe that we fully addressed the concerns raised by Reviewer 3 in the revised manuscript and in our detailed response provided below. At the same time, we will be more than happy to provide additional clarifications and to introduce extra modifications to the manuscript should Reviewer 3 have any additional comments, questions or concerns.
>
> In our experiments we have not observed significant difference in performance between the interpretable and the generic model configurations across different datasets. Most of the time, the forecasts of the two models are close and the forecasting errors are strongly correlated. This is clear, for example, from Appendix D1, Tables 11 and 12 (revised manuscript) where we can see that ensembling the interpretable and generic model only results in a marginal overall gain. Thus statistically speaking the outputs of the interpretable and the generic models are equivalent. At the same time, there are plenty of examples in which FORECAST-I and FORECAST-G deviate noticeably. This is expected, because ultimately the two models are different and thus they cannot be expected to produce exactly the same outputs. For example, the interpretable model uses weight sharing across stacks and its basis is constrained to trend in the first stack and Fourier basis in the second one; the generic model does not employ weight sharing and it relies on learnable basis layers. On top of that, even ensembles of models of the same type, e.g. two generic ensembles that were trained on different random batch sequences and different random initializations will have noticeably different, but statistically equivalent outputs. The inductive bias based on trend and seasonality may, of course, reduce the range of time series types for which the interpretable model will be effective. For example, time series with no regular patterns or trends will likely be better handled by the generic model (or an interpretable model of a different kind). At the same time, our empirical results on 3 different datasets containing time series of very different nature imply that the proposed interpretable model is broadly applicable. The range of time series forecasting scenarios where interpretable model based on trend and seasonality is effective appears to be sufficiently broad and includes many cases of practical interest.
>
> In order to address the second question of Reviewer 3, we added Appendix F in the revised manuscript in which we included tables (Tables 17– 22 in the revised manuscript) with the numerical values of time series depicted in Figure 2.  We made sure that the names of signals in Fig. 2 legends and in the table columns match, such that they can easily be cross-referenced. It is clear from Tables 17– 22 that (i) traces STACK1-I and STACK2-I sum up to trace FORECAST-I, (ii) traces STACK1-G andSTACK2-G sum up to trace FORECAST-G, (iii) traces FORECAST-I and FORECAST-G are overall very similar even though their components may significantly differ from each other. This is expected and there is nothing wrong with it. First, it is expected that there may be many pairs of very different curves whose sums look very similar. Second, we conjecture that in a few cases in Fig.2 the perceived difference between -I and -G components may arise due to plot scaling. The small absolute magnitude signals shown at full scale in the plots may create a false impression that they should have a considerable impact on the combined line, but in reality they only contribute level shift. For example, Tables 19 (Fig. 2, row 3, Monthly) and 22 (Fig.2, row 6, Hourly) may be good illustrations of this behaviour. We hope that the tables with numerical values help to clarify this matter.

---

### Official Review · AnonReviewer2 · 2019-10-28
**Official Blind Review #2**

**Rating:** 6

**Review:**

The paper proposes a DL architecture that achieves better performance on time series prediction. The proposed architecture is relatively straightforward and composes residual blocks. While the paper does achieve superior results, a lot of the text is devoted to comparing to prior work and arguing that DL approaches can do better than hand-crafted approaches, instead of focussing on the importance of specific technical contributions made in the paper.

My main concerns are:
(a) The main technical idea in this paper is the use of back-casting and forecasting (i.e. doubly residual connections). However, no ablations are provided to show how important doubly residual connections are. What is the performance, if the DL architecture is kept exactly the same, except for:
    (i) No residual connections in backcasting (simply feed in the overall input to every block)
    (ii) In addition to (i), make no backcasting predictions

(b) Given no architectural details of the previous ML methods are provided, its unclear if the current architecture is better because it has more parameters or it is indeed the doubly residual idea that is important.

(c) Authors make a point about interpretability — but interpretability is only achieved using domain specific knowledge. I am not really sure, what is so novel about this.

Overall, this seems to be a good applications paper which has been optimized for performance. As a research paper, the contributions are less clear. Answers to my concerns above will help clarify. Given my current concerns, I cannot recommend this paper to be accepted.

--------------------------
Post Rebuttal:

Authors have addressed several of my concerns by providing detailed ablations and the paper reads much better. I still have some concerns:
(a) Interpretability in this work is obtained by using standard basis functions used by the TS community. Such basis functions can be used with any architecture, not just the doubly residual connection networks. Also, its common for people to make use of domain specific knowledge, for e.g. predicting spectrograms when analyzing audio data. In these cases, yes predictions are interpretable. Not sure, what is the contribution that authors are claiming.

(b) The connection to meta-learning seems to be a bit mis-leading. The authors claim that all the weights of the network form the outer loop and the stage-wise predictions of \theta corresponds to the inner loop. Can I simply not think of \theta as features? In that case, this logic will apply to any deep neural network. Not sure what exactly authors want to say. Under this view all deep networks are doing meta-learning. I might have misunderstood something here -- please correct me, if so.

I am changing my rating to weal accept, but I also hope the authors will address the concerns outlined above.


**Experience Assessment:**

I have read many papers in this area.

**Review Assessment: Checking Correctness Of Derivations And Theory:**

N/A

**Review Assessment: Checking Correctness Of Experiments:**

I carefully checked the experiments.

**Review Assessment: Thoroughness In Paper Reading:**

I read the paper at least twice and used my best judgement in assessing the paper.

---

> ### Author Response · Authors · 2019-11-05
> **Details of experimental setup**
>
> We would like to sincerely thank Reviewer 2 for the thorough analysis of the manuscript and for insightful feedback. We are committed to fully address the comments raised by the reviewer, including conducting the additional experiments. Our detailed response will follow as soon as the experimental results are obtained. In the meantime, we would like to ask the reviewer to kindly clarify the desired experimental setup. We have the following interpretations of the experimental setup and we would like to confirm which of our interpretations is correct. We apologize for any misunderstanding and we hope that Reviewer 2 may provide necessary clarifications.
>
> **Interpretation 1**
> We interpret the setup in point (a) (i) as follows: Residual connections are disabled, and all blocks are fed the same overall input. We assume that the overall input refers to the model input. The blocks then forecast in parallel and their individual outputs are summed to make the final forecast.
> In this interpretation, the setup described in point (a) (ii) is unclear to us: (a) (i) already seems to imply that the backcast predictions are not made.
>
> **Interpretation 2**
> We interpret the setup in point (a) (i) as follows: residual connections are disabled and the backcast connection of the previous block is fed as the input to the next block. The model input is only fed in the first block.
> In this interpretation, (a) (i) implies that the backcast connections are the only links between successive blocks. So, we interpret (a) (ii) as follows: the backcast connections are not made. Instead, the overall model input is fed into each block. The blocks then forecast in parallel and their individual outputs are summed to make the final forecast.

---

> ### Author Response · Authors · 2019-11-08
> **Ablation studies addressing concerns (a) and (b)**
>
> We addressed the concerns (a) and (b) raised by Reviewer 2 by conducting an additional ablation study to confirm the effectiveness of the proposed doubly residual topology. The results of the ablation experiment are summarized in Appendix C.3 DOUBLY RESIDUAL STACKING.
>
> We considered the following alternatives to the original architecture (N-BEATS-DRESS) proposed in the original draft.
>
> PARALLEL is depicted in Fig. 4b. This is the alternative where the backward residual connection is
> disabled and the overall model input is fed to every block. The blocks then forecast in parallel using
> the same input and their individual outputs are summed to make the final forecast.
>
> NO-RESIDUAL is depicted in Fig. 4c. This is the alternative where the backward residual connection
> is disabled. Unlike PARALLEL, in this case the backcast forecast of the previous block is fed as input
> to the next block. Unlike the usual feed-forward network, in the NO-RESIDUAL architecture, each
> block makes a partial forecast and their individual outputs are summed to make the final forecast.
>
> LAST-FORWARD is depicted in Fig. 4d. This is the alternative where the backward residual
> connection is active, however the model level forecast is derived only from the last block. So, the
> partial forward forecasts are disabled. This is the architecture that is closest to the classical residual
> network.
>
> NO-RESIDUAL-LAST-FORWARD is depicted in Fig. 4e. This is the alternative where both
> backward residual and the partial forward connections are disabled. This is therefore a simple
> feed-forward network, but very deep.
>
> Tables 7-10 in Appendix C.3 present the quantitative results comparing these architectural alternatives to the proposed N-BEATS configuration (N-BEATS-DRESS).  The results demonstrate that the doubly residual stacking topology provides a clear overall advantage over the alternative architectures in which backcast residual links or the partial forward forecast links or both of those are disabled.
>
> We highly value the time and effort Reviewer 2 has invested in reviewing the draft and we would sincerely appreciate additional feedback from Reviewer 2 regarding the revised version of the paper including additional ablation studies.

---

> ### Author Response · Authors · 2019-11-13
> **Response to Reviewer 2**
>
> First of all, we would like to sincerely thank Reviewer 2 again for providing a thorough insightful review. We believe that we fully addressed the concerns raised by the Reviewer in the revised manuscript and in our detailed responses below. At the same time, we will be happy to address any additional questions and concerns that Reviewer 2 may have. We briefly summarize our previous message for clarity (please refer to the message dated 08 Nov 2019 for more details). To address (a,b) we conducted an additional ablation study to confirm the effectiveness of the doubly residual topology with respect to several alternatives. We included the results of the study in Appendix C.3, Tables 7-10. These results demonstrate the effectiveness of the proposed doubly residual topology proving that the proposed architecture provides gain that cannot be achieved by simply increasing the number of parameters.
>
> Looking at this from another angle and to additionally address (b): the proposed architecture allows to more efficiently use a fixed number of parameters. The conclusion is supported by the ablation study presented in the original manuscript. In Table 6, rows 3 and 4, the interpretable architecture is compared in configurations with 1 and 3 blocks per stack, respectively. Since the interpretable architecture shares all the parameters in blocks within a stack, the number of parameters in configurations 1 block per stack and 3 blocks per stack is actually the same! Yet, stacking blocks with exactly the same parameters turns out to be beneficial. We believe this is an interesting finding in itself and this is what we alluded to in Appendix A discussing the connections of N-BEATS to meta-learning paradigm.
>
> To address the concern raised by the Reviewer 2 in (c) we would like to point out that the proposed approach to interpretability is novel, because (i) we propose a novel generic architectural approach that is very flexible and it can operate both in interpretable and non-interpretable regimes, (ii) we devise a concrete novel recipe on exactly how the proposed architecture can be configured to make its outputs interpretable and (iii) we empirically demonstrate that both generic and interpretable configurations have comparable performance across 3 different datasets.
>
> Additionally, Reviewer 2 characterized our paper as "applications paper which has been optimized for performance". We believe that our paper proposes a novel architecture and is focused on proving that it is generically applicable to solving TS forecasting, rather than on optimizing its performance.  Concretely, our results are based on 3 different well established datasets. The datasets contain time series (TS) of very different nature and the size of datasets is drastically different (100,000; 3003 and 1311 time series). We applied N-BEATS to the datasets in two different configurations (interpretable and generic) and we kept all the architectural hyperparameters (number of blocks, layer width, number of layers, etc.) fixed at the same values (please refer to Table 12) across datasets and datasets' different subsets, varying significantly in their properties. N-BEATS still achieved SOTA on all of the datasets under these constraints. Since the submission of the manuscript we have significantly extended the empirical results by applying N-BEATS to 2 more datasets in the same architectural configuration and again achieved SOTA. We have not added those results to the revision to avoid creating impression that we add irrelevant material instead of focusing on addressing the points raised by Reviewers. However, we will gladly add those to a new revision if we are explicitly instructed to do so by Reviewers or Area Chairs.
>
> Finally, we would like to address the point about spending a lot of space to compare to prior work. The mission of our paper is to help to build a bridge between ML and TS forecasting. One of the objectives of our study is to fix a long standing problem pointed out by Makridakis et al., 2018a: "... we hope that those in the field of AI and ML will accept the empirical findings and work to improve the forecasting accuracy of their methods. A problem with the academic ML forecasting literature is that the majority of published studies provide forecasts and claim satisfactory accuracies without comparing them with simple statistical methods or even naive benchmarks." To address it, we proposed a novel deep architecture, we used the datasets and error metrics established by the TS community and we convincingly demonstrated that ML is a promising tool for solving TS forecasting problems. On the other hand, in addition to the novel architecture, we expose the ML community to an important yet underrated problem of TS forecasting whose business impact ranges in millions of dollars per percent of accuracy (Kahn, 2003), to new datasets and a more rigorous evaluation methodology based on comparison to all known prior art, including classical models.

---

> > ### Public Comment · ~MENGYUE_ZHA1 · 2020-03-27
> > **Questions on the backcast mechanism**
> >
> > I am a student currently reading this paper and I do have some questions on the backcast mechanism. I understand your idea of feeding the next block i+1 by the residual x_i - x_i^hat produced by block i. However, why it’s necessary to use x_i - x_i^hat as the residual rather than y_i - y_i^hat. After all, the latter one is more common in previous studies in other sub fields like boosting or pure statistical time series. Besides, creating a x_i^hat purposely must introduce more parameters and complexity of computation. Could the gain of choosing x_i -x_i^hat compensate for that? Another concern is that in order to make sure your essential idea of backcast topology work, you have to guarantee that x_i^hat and y_i^hat have some kind of correspondence. How did you do that? I know the ‘forecasting materials’ theta_i^f and theta_i^b are based on the same h_i,4. This could be regard as a kind of correspondence between x_i^hat and y_i^hat. Whereas, more strict mathematically, you should make sure that the linear_i^f and linear_i^b used to map h_i,4 must be the ‘inverse’ to each other. ‘Inverse’ is not the strict one used in linear algebra. Just wanna say that your paper seems have no mechanism to make sure x_i^hat is fully correspondent to y_i^hat.  I read your reply to reviewer#2 and it seems you haven’t done an experiment using y_i-y_i^hat as the residual rather than x_i-x_i^hat. The idea of creating a x_i^hat to figure out what has been digested by block i and then feed the leftovers to block i+1 looks fancy. Intuitively, I believe this setting works but I really need your further explanation to convince myself. Thanks!

---

### Decision · Program_Chairs · 2019-12-19

**Decision:**

Accept (Poster)

**Comment:**

The paper received positive recommendation from all reviewers. Accept.